# A GEOMETRIC LENS ON LLM ABILITIES THROUGH JOINT EMBEDDING ITEM RESPONSE THEORY

## ABSTRACT

Standard LLM evaluation practices compress diverse abilities into single scores, obscuring their inherently multidimensional nature. We present **JE-IRT**, a geometric item-response framework that embeds both LLMs and questions in a shared space. For question embeddings, the *direction* encodes semantics and the *norm* encodes difficulty, while correctness on each question is determined by the geometric interaction between the model and question embeddings. This geometry replaces a global ranking of LLMs with topical specialization and enables smooth variation across related questions. Building on this framework, our experimental results reveal that out-of-distribution behavior can be explained through directional alignment, and that larger norms consistently indicate harder questions. Moreover, JE-IRT naturally supports generalization: once the space is learned, new LLMs are added by fitting a single embedding. The learned space further reveals an LLM-internal taxonomy that only partially aligns with human-defined subject categories. JE-IRT thus establishes a unified and interpretable geometric lens that connects LLM abilities with the structure of questions, offering a distinctive perspective on model evaluation and generalization.

## 1 INTRODUCTION

Large language models (LLMs) have advanced rapidly in both capability and diversity, with new models released at an accelerating pace (Guo et al., 2025; Yang et al., 2025). Evaluating these models has become a central activity, typically relying on benchmark datasets and reporting performance in the form of accuracy, aggregate scores, or leaderboard rankings (Hendrycks et al., 2021; Srivastava et al., 2023). Such evaluations provide a convenient summary and allow models to be compared at scale. Beyond aggregate scores, other approaches include human alignment (Ji et al., 2023; Kirk et al., 2024), which assesses whether model outputs match human preferences, and profiling, which highlights model strengths and weaknesses (Liang et al., 2023; Perez et al., 2023). Profiling methods often group benchmarks into subject categories and measure relative ability across them (Bisk et al., 2020; Srivastava et al., 2023).

While these practices are useful, they capture only part of the picture. Aggregate scores compress heterogeneous abilities into a single number, even though LLM competence is multidimensional (Liang et al., 2023). As for profiling, human-defined subjects in benchmarks reflect educational curricula, whereas LLMs are optimized on mixed-domain corpora with objectives that do not encode explicit subject boundaries. Rather than relying solely on curricular labels, we turn to how models themselves separate and relate questions in their internal representation.

We adopt a complementary perspective that focuses directly on the *interaction* between models and questions. The interaction has two sides: how a given model responds to an individual item, and how items are organized when viewed through the lens of the models. Studying this interaction enables item-level prediction of LLM behavior and reveals how subjects are organized in the model's representational geometry. This capability of item-level prediction is also practical—for instance, routing systems benefit from knowing which model may succeed on which query (Ong et al., 2025; Gururangan et al., 2022).

Guided by this perspective, a natural approach would be to simulate interactions through a model's *ability* score and an question's *difficulty* score, as in traditional item response theory (IRT) (Hambleton & Swaminathan, 2013). However, two empirical patterns challenge this view. First, a single

scalar ability does not induce a universal ordering of LLMs across items: traditional item response models fail to recover a consistent monotone relationship between ability and correctness within benchmarks. Second, behavior varies smoothly across semantically related items: when questions are similar, predicted responses should remain close. Motivated by these observations, we seek a formulation in which these two empirical patterns are captured by the interaction between ability and difficulty.

To explore this idea, we propose **JE-IRT**, a framework that combines joint embedding learning with ideas from item response theory. In JE-IRT, both models and questions are embedded in a shared space; their geometric interaction determines the probability of a correct response. Crucially, *direction* captures semantic specialization (what a question is about), and *norm* reflects difficulty (how hard it is). Unlike traditional IRT, which assigns content-agnostic item parameters learned only from response patterns, JE-IRT ties difficulty and discrimination to question content and does not assume a scalar ability that orders all models. Empirically, JE-IRT provides accurate item-level predictions and scales to new models: after learning question embeddings once, a new LLM can be integrated by training only its embedding, reaching near-joint-training performance with a small fraction of its data. The learned geometry is interpretable: directions organize topical specialization, norms track difficulty, and clustering reveals an LLM-centric taxonomy that only partially aligns with human-defined subjects.

We highlight three main contributions of this work:

- Using traditional item response theory (IRT) and analysis of LLM responses, we show that the assumed total order between model ability and question difficulty does not hold, even within fine-grained benchmarks. This calls for moving beyond scalar rankings toward frameworks that capture richer model–question interactions.

- We propose **JE-IRT**, a geometric IRT formulation where question *orientation* captures semantics and question *norm* captures difficulty, and correctness is driven by projected ability minus norm. JE-IRT outperforms baselines in prediction accuracy and is scalable, as new LLMs can be added through lightweight embedding fine-tuning instead of full retraining.

- We validate the geometry by showing that directional alignment predicts out-of-distribution drops, and norms reliably indicate difficulty. We also show that clustering the embeddings reveals subject divisions that differ from human-defined categories, suggesting that LLMs organize knowledge in their own representational sense.

## 2 RELATED WORK

**Item Response Theory in NLP and ML.** Item Response Theory (IRT) has been adapted from educational testing to NLP and machine learning as a way to evaluate datasets and models beyond aggregate accuracy (Downing, 2003; Lalor et al., 2019; Cheng et al., 2019; Rodriguez et al., 2021b). By modeling item difficulty and discrimination alongside latent model ability, IRT reveals variation that standard metrics obscure. Early work introduced IRT-calibrated test scales for NLP (Lalor et al., 2016), followed by applications to benchmark quality, dataset bias, and model profiling (Rodriguez et al., 2021a; Bachmann et al., 2024). More recently, extensions have generalized IRT with multidimensional and neural variants to capture the diverse abilities and large-scale evaluations of modern LLMs (Varadarajan et al., 2024; Zhou et al., 2025), and adapted it for adaptive testing in pretraining (Hofmann et al., 2025). Unlike these approaches, which assign parameters to each question, our framework learns difficulty and discrimination directly from question semantics, enabling interpretable analysis that scales beyond parameter-per-question formulations.

**Representation-Based Understanding and Evaluation.** Representation-based approaches embed models and questions into a shared space, offering a geometric view of their interactions and exposing latent capabilities. This line of work is still rare: EmbedLLM (Zhuang et al., 2025) explicitly learns embeddings that predict correctness and reveal model specialization, while IRT-Router (Song et al., 2025) leverages multidimensional IRT for routing, using latent representations only as an implicit means to optimize query assignment. In contrast, our proposed JE-IRT makes the geometry explicit: embedding norm reflects difficulty and direction captures semantics, yielding an interpretable representation of model–task interactions. Evaluation and routing follow naturally, but the central aim is understanding.

## 3 THEORETICAL FOUNDATIONS AND OUR FRAMEWORK

**Traditional Item Response Theory.** In traditional item response theory, typically expressed through the 2-parameter logistic (2PL) model, each LLM $M_i$ is assigned an ability score $\theta_i$, and each question $Q_j$ is associated with a discrimination score $a_j$ and a difficulty score $b_j$. The ability score reflects the overall competence of the model, the difficulty score specifies how challenging a question is, and the discrimination score measures how effectively the question separates strong models from weak ones. Note that in the 2PL model, discrimination $a_j$ and difficulty $b_j$ are content-agnostic latents: they are inferred solely from response patterns—whether each model answered each question correctly—rather than from the question's text or semantics. Given $\theta_i$, $a_j$, and $b_j$, the probability that model $M_i$ answers question $Q_j$ correctly is given by

$$P(M_i, Q_j) = \sigma\left(a_j(\theta_i - b_j)\right), \tag{1}$$

where $\sigma$ denotes the sigmoid function. A key assumption in traditional IRT is the global ordering assumption: models with higher ability scores are expected to have a higher probability of answering every question correctly. This is enforced by requiring discrimination parameter $a_j$ to be non-negative, since otherwise a lower-ability model could have a higher probability of correctly answering a difficult question. However, as we demonstrate later in Section 4.1, this assumption is often violated, undermining the usefulness of the formalism.

**JE-IRT.** To address the limitations of traditional item response theory, we propose the *Joint Embedding Item Response Theory* (JE-IRT). Instead of assigning scalar ability and difficulty scores, we embed both models and questions in a shared higher-dimensional space $\mathbb{R}^d$, denoted by $\mathbf{E}_{M_i}$ for model $M_i$ and $\mathbf{E}_{Q_j}$ for question $Q_j$. We define the ability of model $M_i$ on question $Q_j$ as

$$\Theta_{M_i, Q_j} = \frac{\mathbf{E}_{Q_j} \cdot \mathbf{E}_{M_i}}{\|\mathbf{E}_{Q_j}\|}, \tag{2}$$

which corresponds to the length of the projection of the model embedding onto the direction of the question embedding. The probability of a correct response is then given by

$$P(M_i, Q_j) = \sigma\left(\Theta_{M_i, Q_j} - \|\mathbf{E}_{Q_j}\|\right). \tag{3}$$

In this formulation, we aim to disentangle the question's difficulty and topical focus into the magnitude and direction of its embedding respectively. This structure yields an important property: when a model's projected ability is large along the direction of a question's embedding, it is more likely to answer that question correctly, while the question embedding norm governs how difficult the question is overall.

**Proposition 1 (No Global Ability Ordering in JE-IRT)** *Under the JE-IRT framework, there exist models $M_1$, $M_2$ and questions $Q_1$, $Q_2$ such that*

$$\Theta_{M_1, Q_1} > \Theta_{M_2, Q_1} \quad but \quad \Theta_{M_2, Q_2} > \Theta_{M_1, Q_2},$$

*where $\Theta_{M_i, Q_j}$ denotes the ability score of model $M_i$ on question $Q_j$.*

The proof can be found in Appendix C. This proposition highlights that JE-IRT naturally supports modeling specialized abilities, where no global ability ordering is assumed or enforced across all questions.

**Proposition 2 (Stability of Predicted Probabilities Under Similar Questions)** *Let $M$ be a LLM with embedding $\mathbf{E}_M$, and let $Q_1, Q_2$ be two questions with embeddings $\mathbf{E}_{Q_1}, \mathbf{E}_{Q_2}$. Assume*

$$\cos(\mathbf{E}_{Q_1}, \mathbf{E}_{Q_2}) = 1 - \varepsilon$$

*for some $\varepsilon > 0$, and define $P(M, Q) = \sigma\left(\Theta_{M, Q} - \|\mathbf{E}_Q\|\right)$ as in Eq. (3). Then*

$$|P(M, Q_1) - P(M, Q_2)| \leq \tfrac{1}{4}\left(\sqrt{2\varepsilon}\,\|\mathbf{E}_M\| + \left|\|\mathbf{E}_{Q_1}\| - \|\mathbf{E}_{Q_2}\|\right|\right). \tag{4}$$

*In particular, if $\|\mathbf{E}_{Q_1}\| = \|\mathbf{E}_{Q_2}\|$, then $\left|P(M, Q_1) - P(M, Q_2)\right| \leq \tfrac{1}{4}\sqrt{2\varepsilon}\,\|\mathbf{E}_M\|$.*

The proof is provided in Appendix E. This result shows that predicted probabilities change only slightly when two questions are similar in geometry. Specifically, the prediction gap for any model is bounded by two terms: an angular separation term (capturing semantic misalignment) and a norm difference term (capturing difficulty gap). In the special case where the norms match, the gap depends solely on angular alignment. Together, these two propositions capture the guarantees that JE-IRT is designed to provide: it can represent specialized abilities without enforcing a global ordering, while also preserving consistent probability estimates across semantically related questions.

**Model Structure.** For the framework to generalize to unseen questions, the question embeddings must be determined by the question's content. That is, instead of treating the embedding as a free parameter fixed to the training set, we define it as a function of the question itself:

$$\mathbf{E}_{Q_j} = f_\theta(Q_j), \tag{5}$$

where $f_\theta(\cdot)$ is a learned function that maps the question content to its embedding. This allows us to evaluate the model and test the theory on unseen questions and allows the model to be used for downstream tasks.

Our framework consists of three components: a frozen base encoder $f_{\text{base}}$, a trainable adapter network $g_\theta$, and an embedding table $\mathbf{T}$ for LLMs. Given a question $Q$, it is successively processed by the base encoder and the adapter network, yielding the question embedding $\mathbf{E}_Q = g_\theta(f_{\text{base}}(Q))$. The adapter $g_\theta$ is implemented as a two-layer MLP with hidden size twice the base encoder output dimension, enabling it to disentangle features from the frozen representation before projection into the final embedding space.

Each LLM is associated with a learnable embedding vector $\mathbf{E}_M = \mathbf{T}[M]$. Thus, the trainable parameters of the framework are the adapter weights $\theta$ and the embedding table $\mathbf{T}$. The predicted correctness probability is then computed from Eq. (3), with training carried out using binary cross-entropy against ground truth labels.

# 4 EXPERIMENTS AND ANALYSIS

We present our empirical results in this section. All experiments are conducted on the EmbedLLM correctness dataset (Zhuang et al., 2025). This dataset includes evaluation results for 112 models across 10 benchmarks and, to the best of our knowledge, represents the largest collection of LLMs evaluated on a shared set of questions.

## 4.1 NO UNIVERSAL TOTAL ORDERING ACROSS MODELS

A common assumption in traditional IRT is that a single scalar *ability* induces a *global* ranking of models: if $M_a$ is more able than $M_b$ (e.g., $\theta_a > \theta_b$ in Eq. (1)), then for *every* item $Q_j$ we must have $P(M_a, Q_j) \geq P(M_b, Q_j)$. Equivalently, under the 2PL with strictly positive discriminations $(a_j > 0)$, the item-characteristic curve (ICC) is monotone increasing in ability, so a single total order should explain correctness across items. We show this assumption is violated for LLMs.

**Evidence from fitting the 2PL.** We fit the 2PL in Eq. (1) to our data without enforcing nonnegativity on $a_j$. Figure 1 summarizes the result. *Left:* each dot is a question positioned by its estimated difficulty $b_j$ and discrimination $a_j$. Many items lie at $a_j \leq 0$ and a substantial mass has $a_j \approx 0$, contradicting the monotone-with-ability premise. *Right:* four representative items illustrate the failure modes: (i) negative discrimination $(a_j < 0)$, (ii) near-zero discrimination $(a_j \approx 0)$, and (iii–iv) ostensibly positive discrimination $(a_j > 0)$. Each small plot is an item-characteristic curve (ICC) for a single question: the blue curve is the 2PL prediction as a function of model ability, and the black dots mark the observed correctness of all LLMs at their fitted abilities. Even when $a_j > 0$, the empirical points do *not* align monotonically with ability—many lower-ability models answer correctly while some higher-ability models fail. When $a_j \approx 0$, the ICC is essentially flat and captures no relationship at all. Moreover, for 49% of items (1,275/3,000), the fitted 2PL saturates to near-unanimous predictions (all correct or all incorrect), whereas the *actual* data are unanimous on only 2.7% of items (80/3,000). Taken together, the parameter scatter and the four ICCs in Figure 1 show that a meaningful global order doesn't exist and a single scalar ability fails to capture the real ability of LLMs.

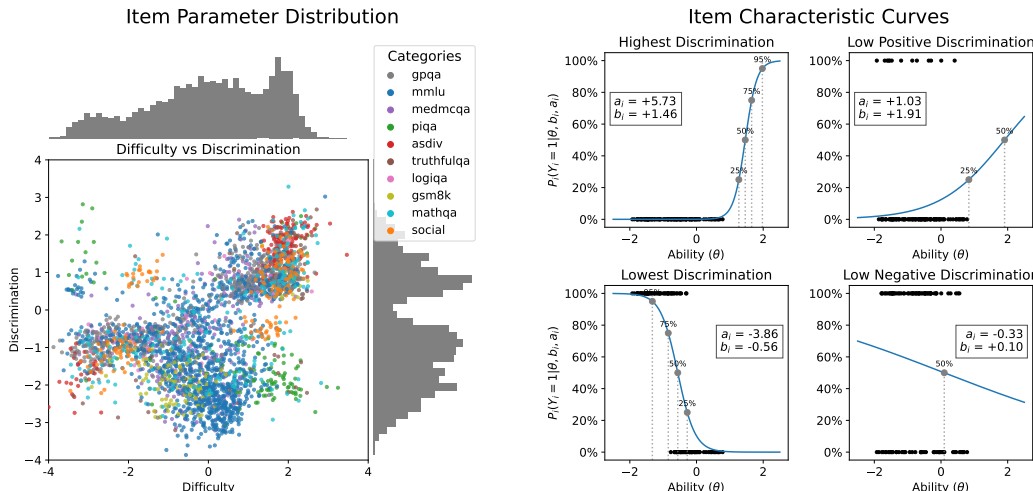

Figure 1: Traditional 2-parameter logistic (2PL) IRT assumptions fail on LLM data. *Left:* Estimated difficulty $b_j$ (x-axis) and discrimination $a_j$ (y-axis) for each question from a 2PL fit on the EmbedLLM test set. Many items have $a_j \leq 0$ or $a_j \approx 0$, contradicting the monotone-with-ability premise behind a universal total order. *Right:* Four example questions (negative, near-zero, and two positive $a_j$); each panel overlays the fitted item-characteristic curve (ICC) in blue with empirical outcomes of all models (black dots at their fitted abilities $\theta$). Even when $a_j > 0$, correctness is not monotone in $\theta$; when $a_j \approx 0$, the ICC is essentially flat. Together, the panels illustrate why a single scalar ability cannot explain these data.

**Evidence from correct-set inclusion.** A strict global order has a concrete set-theoretic implication: if $M_j$ is "stronger" (higher overall accuracy) than $M_i$, then the set of questions it answers correctly should *superset* the weaker model's correct set. Let $Q(M)$ be the questions a model answers correctly and define

$$R(M_i, M_j) \ = \ \frac{|\,Q(M_i) \setminus Q(M_j)\,|}{|\,Q(M_i)\,|}.$$

If a total order held, $R(M_i, M_j) \approx 0$ for all weaker–stronger pairs. In practice, heat maps on MATHQA and GSM8K (Figs. 13, 14 in Appendix O) show widespread, non-trivial values across many pairs: nominally stronger models routinely *miss* items solved by weaker ones. This violates set inclusion and independently rules out a universal total order within benchmarks.

Both perspectives—parametric (2PL fits) and set-theoretic (correct-set inclusion)—converge: LLM abilities do not admit a single, benchmark-wide scalar ranking. This motivates our JE-IRT design, which replaces a universal order with a geometric interaction that allows specialization across semantic *directions* and difficulty *norms*, matching the item-level diversity observed in the data.

## 4.2 PERFORMANCE EVALUATION OF JE-IRT

We evaluate the learned geometry by predicting, at question level, whether an LLM answers correctly. JE-IRT is trained with two frozen base encoders—ModernBERT-Large (Warner et al., 2025) and all-mpnet-base-v2 (Reimers & Gurevych, 2019)—across a range of embedding dimensions. For each embedding dimension $d$, we select the checkpoint with the lowest validation loss and report test-set performance.

**Accuracy vs. dimension.** As the left panel of Figure 2 shows, test accuracy increases with $d$ for both encoders and peaks around $d = 256$, indicating that the predictive structure JE-IRT needs is effectively low-dimensional. For comparison, we include the EmbedLLM baseline, which uses a fixed 252-dimensional embedding per model. JE-IRT surpasses this baseline with far smaller embeddings—at just $d = 16$ with ModernBERT and around $d = 64$ with the sentence-transformer—yielding an order-of-magnitude (or multi-fold) reduction in dimensionality while matching or exceeding ac-

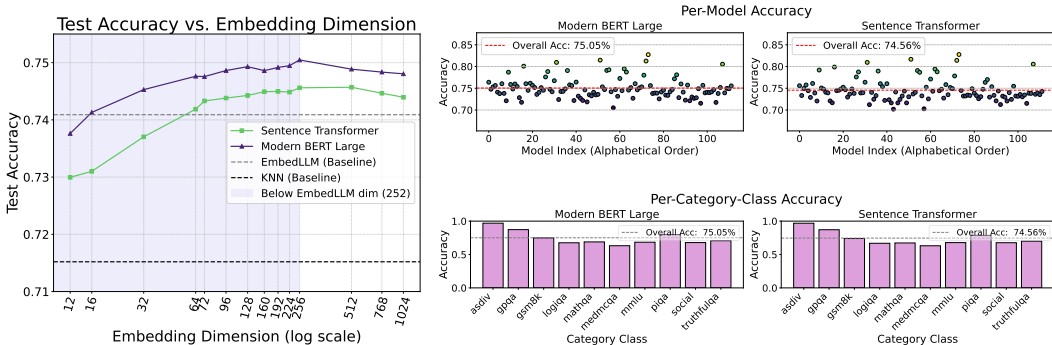

Figure 2: (Left) Test accuracy as a function of embedding dimension (log scale) for two base encoders. Each point shows the best-performing model at each dimension, selected by validation loss. The dashed black line denotes the baseline performance from EmbedLLM (252-dimensional embedding), and the gray region highlights lower-dimensional regimes. (Top right) Accuracy breakdown across individual LLMs for models trained with 256-dimensional embeddings. (Bottom right) Accuracy by benchmark category for the same 256-dimensional models.

curacy. This demonstrates that JE-IRT captures the essential interaction between models and questions with substantially fewer parameters.

**Per-model and per-benchmark breakdowns.** Focusing on the best 256-dimensional setting, the top-right plot reports per-model accuracies (one point per LLM; dashed line = overall mean), and the bottom-right plot aggregates the same models by benchmark category. The distributions are tight around the mean along both axes: gains are not driven by a handful of outlier models, nor are they confined to a particular benchmark. Instead, performance is consistent across the portfolio of 112 LLMs and the 10 benchmarks, indicating that the learned embeddings generalize beyond specific models or tasks and reflect broad, reusable patterns of question difficulty and model capability.

These results underscore the strong performance of the framework, demonstrating that its geometric inductive bias effectively captures key interactions between LLMs and questions. We further investigate this geometry in Section 4.4 and Section 4.5, where we analyze orientation and magnitude separately.

## 4.3 DATA-EFFICIENT SCALABILITY TO NEW LLMS

Given the rapid pace at which new models appear, a practical evaluation framework should let us onboard additional LLMs without retraining everything. We show that JE-IRT is scalable by demonstrating that it can incorporate each new model using a *single* embedding $E_M \in \mathbb{R}^d$ while keeping the question embeddings frozen. To assess scalability, we consider two settings: leave-one-out experiments on the EmbedLLM dataset and evaluation on more recent LLMs.

**Leave-one-out on EmbedLLM.** We randomly sample 10 models from the full set of 112. For each run, one model is excluded during training, and JE–IRT is trained on the remaining 111. We then simulate adding the excluded model by freezing the question embeddings and fine-tuning only its model embedding. To assess data efficiency, we subsample $1\%$–$100\%$ of the training set and report test accuracy in Table 1. Results are averaged over the 10 randomly selected LLMs (listed in Appendix M). The *All* column shows the accuracy of the same 10 LLMs when all 112 models are trained together. Performance plateaus quickly: using only $10\%$ of the held-out model's data, accuracy is already within $0.5\%$ of the ALL setting for both base encoders.

**Adding recent LLMs.** We extend the analysis by *adding* six more recent LLMs such as Qwen3-30B-A3B (Qwen Team, 2025) and evaluating across eight benchmarks (details in Appendix N). As before, we freeze question embeddings and train only the new models' embeddings, with the same $1\%$–$100\%$ subsampling protocol. Table 2 shows the experimental results. With the sentence-transformer encoder, accuracy remains high even with very limited data; with ModernBERT, accu-

racy degrades more with a limited training data, but maintains strong performance with $40\%$ of the data. These results indicate that JE-IRT remains effective when onboarding up-to-date models.

This efficiency can be explained by the fact that, once question embeddings are fixed, fitting a new model embedding reduces to *a logistic regression problem with d parameters*.[1] Since logistic regression has sample complexity that scales linearly with the dimension (Ng & Jordan, 2001; Shalev-Shwartz & Ben-David, 2014), this naturally explains the fast plateaus in Table 1–2. Empirically, the fact that tuning only $E_M$ nearly matches joint training suggests the learned question geometry is stable and transferable across models, underscoring the scalability of the framework.

Table 1: Test accuracy (%) of 10 held-out LLMs integrated into a trained JE-IRT model with different fractions of their training data (1%–100%). Results are averaged over the 10 models with subscripts as standard deviations. The "All" column reports accuracy when trained with all models.

| Encoder | 1% | 5% | 10% | 20% | 40% | 60% | 80% | 100% | All |
|---|---|---|---|---|---|---|---|---|---|
| **Modern BERT** | $71.03_{\pm3.93}$ | $73.86_{\pm2.14}$ | $74.18_{\pm2.16}$ | $74.35_{\pm2.07}$ | $74.35_{\pm2.10}$ | $74.35_{\pm2.11}$ | $74.39_{\pm2.11}$ | $74.30_{\pm2.15}$ | $74.69_{\pm2.07}$ |
| **Sent-Trans** | $72.30_{\pm2.87}$ | $73.63_{\pm2.43}$ | $73.60_{\pm2.37}$ | $73.78_{\pm2.34}$ | $73.74_{\pm2.31}$ | $73.81_{\pm2.39}$ | $73.71_{\pm2.44}$ | $73.81_{\pm2.42}$ | $73.85_{\pm2.42}$ |

Table 2: Test accuracy (%) of 6 up-to-date LLMs integrated into a trained JE-IRT model with different fractions of their training data (1%–100%). Results are averaged over the 6 models.

| Encoder | 1% | 5% | 10% | 20% | 40% | 60% | 80% | 100% |
|---|---|---|---|---|---|---|---|---|
| **Modern BERT** | 63.15 | 65.09 | 68.48 | 70.02 | 71.70 | 72.00 | 72.06 | 72.42 |
| **Sent-Trans** | 73.45 | 73.67 | 74.12 | 74.69 | 75.12 | 75.13 | 75.22 | 75.42 |

## 4.4 OOD GENERALIZATION AND EMBEDDING ORIENTATION

Out-of-distribution (OOD) generalization is central to profiling models, routing tasks, and providing reliable evaluation, since LLMs are inevitably applied beyond benchmark domains. We study OOD behavior from two complementary views: (i) the *traditional* OOD evaluation on held-out benchmarks through a leave-one-out strategy, and (ii) a *geometric* view based on directional alignment in the learned question-embedding space.

**Leave-one-out accuracy on held-out benchmarks.** In each run, one benchmark is excluded from training and used only for evaluation. We compare test accuracy to the model trained on the full dataset in Table 3. Gray cells show the drop relative to full-data training. As expected, holding out a benchmark reduces accuracy, but the magnitude varies markedly by benchmark. MathQA and LogiQA show small drops, suggesting their knowledge domains are well covered by the remaining data, while PIQA and MMLU exhibit larger drops, indicating more benchmark-specific patterns or reasoning skills. We also observe an encoder-dependent effect on GPQA: exclusion causes a substantial drop with the sentence-transformer encoder but a much smaller one with ModernBERT, suggesting the two encoders capture partially different cues even when overall performance is similar.

**Geometric alignment via embedding orientation.** Under JE-IRT, the *direction* of a question embedding reflects its semantics. If a held-out benchmark is directionally aligned with the training pool, Proposition 2 predicts smaller changes in predicted correctness. We quantify alignment with a simple statistic. For a benchmark $\mathcal{B}$, let $u_q := E_q/\|E_q\|$ be the unit direction of question $q$. Define its mean direction

$$\mu_{\mathcal{B}} = \frac{1}{|\mathcal{B}|} \sum_{q \in \mathcal{B}} u_q \quad \text{and} \quad \mu_{\neg\mathcal{B}} = \frac{1}{|\neg\mathcal{B}|} \sum_{q \in \neg\mathcal{B}} u_q,$$

where $\neg\mathcal{B}$ denotes all other benchmarks. The *directional alignment* of $\mathcal{B}$ with the rest is $\cos(\mu_{\mathcal{B}}, \mu_{\neg\mathcal{B}})$. Table 4 reports this measure. While not a perfect predictor, a consistent trend emerges: benchmarks with higher alignment (e.g., LogiQA, MathQA) tend to have smaller leave-one-out drops, whereas lower alignment (e.g., PIQA, GSM8K) is associated with larger declines.

---

[1]See Appendix F for more details.

Table 3: Leave-one-out evaluation results for out-of-distribution (OOD) generalization. Results are reported as accuracy (%) across ten benchmarks. The "Diff" row shows the performance drop when the benchmark is excluded. The last column reports the overall accuracy.

| Encoder | Setting | asdiv | gpqa | gsm8k | logiqa | mathqa | medmcqa | mmlu | piqa | social | truthfulqa | Overall |
|---|---|---|---|---|---|---|---|---|---|---|---|---|
| **Modern BERT** | Leave-Out | 96.26 | 85.08 | 59.05 | 67.14 | 66.11 | 58.02 | 46.50 | 26.67 | 61.19 | 65.67 | 62.80 |
| | All | 96.93 | 87.25 | 74.86 | 67.56 | 68.86 | 63.17 | 68.40 | 79.96 | 67.87 | 70.63 | 75.05 |
| | Diff | 0.67 | 2.17 | 15.81 | 0.42 | 2.75 | 5.15 | 21.90 | 53.29 | 6.68 | 4.96 | 12.24 |
| **Sent-Trans** | Leave-Out | 93.97 | 79.01 | 59.05 | 64.68 | 66.27 | 54.43 | 50.49 | 28.81 | 64.88 | 67.36 | 62.94 |
| | All | 96.86 | 87.11 | 73.94 | 66.82 | 67.30 | 63.09 | 67.84 | 78.66 | 67.67 | 69.92 | 74.56 |
| | Diff | 2.89 | 8.10 | 14.89 | 2.14 | 1.02 | 3.66 | 17.35 | 49.85 | 2.79 | 2.56 | 11.62 |

Table 4: Cosine similarity ($\times 100$) between the mean embedding of each benchmark and the mean embedding of all other benchmarks combined (i.e., leave-one-out). Higher values indicate stronger directional alignment between a benchmark and the rest of the dataset.

| Encoder | asdiv | gpqa | gsm8k | logiqa | mathqa | medmcqa | mmlu | piqa | social | truthfulqa |
|---|---|---|---|---|---|---|---|---|---|---|
| **Modern BERT** | 40.82 | 84.63 | 36.39 | 97.04 | 96.75 | 96.24 | 92.51 | 70.98 | 92.10 | 94.27 |
| **Sent-Trans** | 59.85 | 61.38 | 51.75 | 96.71 | 96.45 | 96.97 | 73.10 | 16.51 | 92.31 | 95.02 |

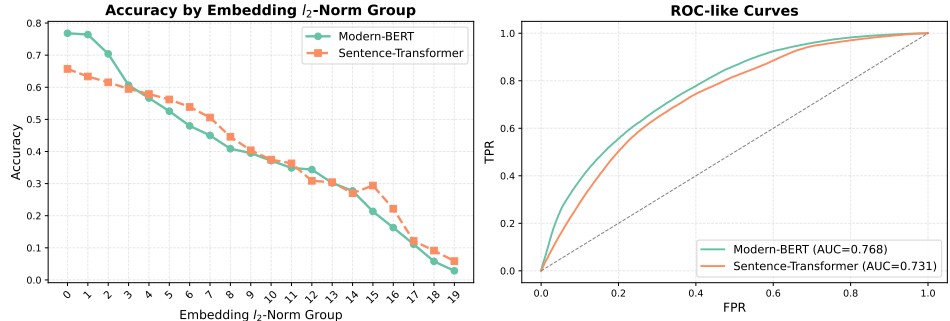

Figure 3: Embedding norm as a proxy for question difficulty. Left: accuracy decreases as the question's embedding norm increases. Right: using $\|\mathbf{E}_Q\|$ as a score and defining *incorrect* as the positive class, ROC-like curves for both encoders achieve AUC 0.73–0.77, confirming that higher norms reliably indicate harder questions.

MMLU is an exception: although it shows moderate cosine similarity with the rest of the benchmarks, its performance drop is substantial. This is likely due to the fact that MMLU contains a broad range of diverse subjects, and its mean direction may not be representative of its true directional coverage.

In summary, through the generalizability study we show that the learned embeddings are robust. Both benchmark-level evaluation and the geometry of embedding orientations provide consistent signals of this robustness, with directional alignment serving as a reliable indicator of how well a model is expected to perform on new or unseen tasks. A more thorough study of the embedding geometry of questions and LLMs can be found in Appendix G.

### 4.5 DIFFICULTIES AND EMBEDDING NORMS

In JE-IRT, the question norm enters the logit subtractively (Eq. (3)): $p(Q) = \sigma\big(u_Q \cdot E_M - \|E_Q\|\big)$. Our inductive bias is that larger $\|E_Q\|$ encodes greater difficulty. We test this with two diagnostics.

**Binned accuracy.** In the left panel of Figure 3, we partition questions into 20 quantile bins by $\|E_Q\|$ and plot average accuracy (averaged over LLMs) per bin. For both base encoders, accuracy

Table 5: Comparison between human-defined subjects and KMeans clustering of learned question embeddings.

| Base Encoders | Purity | Inv. Purity | NMI | Homogeneity | Completeness |
|---|---|---|---|---|---|
| **Modern BERT** | 0.346 | 0.254 | 0.380 | 0.389 | 0.371 |
| **Sent-Trans** | 0.372 | 0.265 | 0.409 | 0.418 | 0.399 |

decreases monotonically (almost always) as $\|E_Q\|$ increases, indicating that higher norms correspond to harder questions.

**ROC-like curves.** In the right panel, we assess how well the embedding norm $\|E_Q\|$ indicates difficulty using ROC-like curves. We use $\|E_Q\|$ as a scalar score and define the positive class as questions answered *incorrectly* (i.e., harder for models). Sweeping a threshold $\tau$ on $\|E_Q\|$, we predict *incorrect* if $\|E_Q\| > \tau$ and *correct* otherwise, then compute the true positive rate and the false positive rate. For both encoders the curves lie well above the diagonal (AUC 0.73–0.77), meaning that, with probability $\approx 0.73$–$0.77$, a randomly chosen incorrect question has larger norm than a randomly chosen correct one (more details in Appendix H).

Taken together, these results show that embedding norms capture question difficulty in a consistent and interpretable way, supporting our intended inductive bias.

### 4.6 HUMAN-DEFINED SUBJECTS VS. LLM-INDUCED SUBJECTS

To compare how LLMs organize questions with human-defined subjects, we focus on MMLU (57 subjects). We cluster the JE-IRT question embeddings into $K = 57$ groups using $k$-means on unit-normalized embeddings, and then compare the induced clusters to the original labels. We report standard agreement measures—purity, inverse purity, NMI, homogeneity, and completeness—in Table 5.

Across metrics, scores fall in a moderate range (roughly 0.25–0.42), indicating partial but not tight alignment between the LLM-induced clusters and human-defined subjects. Two consistent patterns emerge. First, *homogeneity* tends to exceed *completeness*: clusters are internally coherent (questions within a cluster often share a label), but many human-defined subjects are fragmented across multiple clusters. Second, *purity* generally exceeds *inverse purity*, reinforcing the same asymmetry: a cluster often maps cleanly to one dominant subject, yet each subject is spread over several clusters. Together with the moderate *NMI*, this suggests that the model forms a subject taxonomy that is not isomorphic to curricular categories.

This divergence is expected and interpretable: from the model's perspective, items that cluster together require similar *abilities* from the LLM, which need not coincide with textbook subjects. In other words, what humans label as one subject may consist of distinct skills for the model (see Appendix I for an example in which two questions from the same human-defined subject require nearly opposite abilities).

## 5 CONCLUSION AND FUTURE WORK

In this work, we introduced **JE-IRT**, a joint embedding framework for modeling interactions between LLMs and questions. By embedding both into a shared space, JE-IRT enables fine-grained prediction and provides interpretable geometric features that reveal how models generalize and how questions are organized in the LLM's own representational sense. Our experiments demonstrate that JE-IRT learns robust features, scales to new models and benchmarks, and lays a foundation for deeper understanding of LLM behavior.

So far, our analysis has focused on evaluating LLMs' ability to answer questions correctly. In future work, other abilities—such as social behaviors such as honesty, persuasion and deception, or the capacity for emotional support—could also be incorporated into the framework through variants of JE-IRT. Combining different abilities may further reveal hidden connections between seemingly unrelated capabilities and lead to a deeper understanding of LLM behaviors.

REPRODUCIBILITY STATEMENT

The data and code to reproduce our results are publicly available.[2]

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

## A    LLM Usage Disclosure

ChatGPT and Gemini were used for coding assistance (implementation support and debugging) and for polishing language and grammar. All technical ideas, experimental design, analyses, and results are original, and all code was reviewed and verified by the authors.

## B    Limitations

While JE-IRT provides a scalable and interpretable framework for studying LLM–question interactions, its main limitation is that our analysis relies on binary correctness labels. LLMs are inherently probabilistic, and their behavior is better characterized by distributions over responses. Although we did not pursue this direction in our experiments, probabilistic outputs could be incorporated into our framework seamlessly by replacing the binary targets with predicted probabilities in the loss function.

## C    Proof of proposition 1

We prove the existence by giving an example. Considering two questions embeddings $\mathbf{E}_{Q_1}$ and $\mathbf{E}_{Q_2}$. We assume these two embeddings in different directions, i.e.

$$\cos(\mathbf{E}_{Q_1}, \mathbf{E}_{Q_2}) < 1.$$

We can define two LLM embeddings $\mathbf{E}_{M_1}$ and $\mathbf{E}_{M_2}$ as

$$\mathbf{E}_{M_1} = \frac{\mathbf{E}_{Q_1}}{\|\mathbf{E}_{Q_1}\|} \quad \text{and} \quad \mathbf{E}_{M_2} = \frac{\mathbf{E}_{Q_2}}{\|\mathbf{E}_{Q_2}\|}.$$

As a result, we have

$$\Theta_{M_1,Q_1} = \frac{\mathbf{E}_{Q_1} \cdot \mathbf{E}_{M_1}}{\|\mathbf{E}_{Q_1}\|} = 1 > \cos(\mathbf{E}_{Q_1}, \mathbf{E}_{Q_2}) <= \frac{\mathbf{E}_{Q_1} \cdot \mathbf{E}_{M_2}}{\|\mathbf{E}_{Q_1}\|} = \Theta_{M_2,Q_1},$$

$$\Theta_{M_2,Q_2} = \frac{\mathbf{E}_{Q_2} \cdot \mathbf{E}_{M_2}}{\|\mathbf{E}_{Q_2}\|} = 1 > \cos(\mathbf{E}_{Q_1}, \mathbf{E}_{Q_2}) = \frac{\mathbf{E}_{Q_2} \cdot \mathbf{E}_{M_1}}{\|\mathbf{E}_{Q_2}\|} = \Theta_{M_1,Q_2}.$$

This concludes the proof.

## D    Bounded Ability Shift

In this section, we introduce another proposition—a variant of Proposition 2—that emphasizes the underlying geometric structure rather than the prediction outcomes.

**Proposition 3 (Bounded Ability Shift for Similar Questions)** *Let $M$ be a model with embedding vector $\mathbf{E}_M$, and let $\mathbf{E}_{Q_1}$ and $\mathbf{E}_{Q_2}$ be the embeddings of two questions. Suppose*

$$\cos(\mathbf{E}_{Q_1}, \mathbf{E}_{Q_2}) = 1 - \varepsilon$$

*for some $\varepsilon > 0$. Then the difference in ability scores of model $M$ on the two questions is bounded by*

$$|\Theta_{M,Q_1} - \Theta_{M,Q_2}| \leq \sqrt{2\varepsilon}\, \|\mathbf{E}_M\|.$$

Since the direction of question embeddings is intended to encode semantic topics, this bound guarantees that the model's ability remains consistent across semantically similar questions.

*Proof.* Let the ability score of model $M$ on question $i$ be defined as

$$\Theta_{M,i} = \frac{\mathbf{E}_M \cdot \mathbf{E}_{Q_i}}{\|\mathbf{E}_{Q_i}\|}.$$

The difference in ability scores for two questions is

$$|\Theta_{M,1} - \Theta_{M,2}| = \left| \frac{\mathbf{E}_M \cdot \mathbf{E}_{Q_1}}{\|\mathbf{E}_{Q_1}\|} - \frac{\mathbf{E}_M \cdot \mathbf{E}_{Q_2}}{\|\mathbf{E}_{Q_2}\|} \right|.$$

Applying the triangle inequality, we obtain

$$|\Theta_{M,1} - \Theta_{M,2}| = \left| \mathbf{E}_M \cdot \left( \frac{\mathbf{E}_{Q_1}}{\|\mathbf{E}_{Q_1}\|} - \frac{\mathbf{E}_{Q_2}}{\|\mathbf{E}_{Q_2}\|} \right) \right| \le \|\mathbf{E}_M\| \cdot \left\| \frac{\mathbf{E}_{Q_1}}{\|\mathbf{E}_{Q_1}\|} - \frac{\mathbf{E}_{Q_2}}{\|\mathbf{E}_{Q_2}\|} \right\|.$$

Define the normalized question embeddings

$$\hat{\mathbf{E}}_{Q_1} = \frac{\mathbf{E}_{Q_1}}{\|\mathbf{E}_{Q_1}\|}, \qquad \hat{\mathbf{E}}_{Q_2} = \frac{\mathbf{E}_{Q_2}}{\|\mathbf{E}_{Q_2}\|}.$$

By definition of cosine similarity,

$$\cos(\hat{\mathbf{E}}_{Q_1}, \hat{\mathbf{E}}_{Q_2}) = \hat{\mathbf{E}}_{Q_1} \cdot \hat{\mathbf{E}}_{Q_2} = 1 - \varepsilon.$$

Since both $\hat{\mathbf{E}}_{Q_1}$ and $\hat{\mathbf{E}}_{Q_2}$ are unit vectors, we compute

$$\|\hat{\mathbf{E}}_{Q_1} - \hat{\mathbf{E}}_{Q_2}\|^2 = \|\hat{\mathbf{E}}_{Q_1}\|^2 + \|\hat{\mathbf{E}}_{Q_2}\|^2 - 2\hat{\mathbf{E}}_{Q_1} \cdot \hat{\mathbf{E}}_{Q_2} = 2 - 2(1 - \varepsilon) = 2\varepsilon.$$

Thus,

$$\left\| \frac{\mathbf{E}_{Q_1}}{\|\mathbf{E}_{Q_1}\|} - \frac{\mathbf{E}_{Q_2}}{\|\mathbf{E}_{Q_2}\|} \right\| = \|\hat{\mathbf{E}}_{Q_1} - \hat{\mathbf{E}}_{Q_2}\| = \sqrt{2\varepsilon}.$$

Putting everything together, we obtain

$$|\Theta_{M,1} - \Theta_{M,2}| \le \|\mathbf{E}_M\| \cdot \sqrt{2\varepsilon}.$$

# E    PROOF OF PROPOSITION 2

Let $z_i := \Theta_{M,Q_i} - \|\mathbf{E}_{Q_i}\|$ for $i = 1, 2$. By the mean value theorem, there exists $\xi$ between $z_1$ and $z_2$ such that

$$|P(M, Q_1) - P(M, Q_2)| = |\sigma(z_1) - \sigma(z_2)| = \sigma'(\xi)|z_1 - z_2|.$$

Since $\sigma'(t) = \sigma(t)(1 - \sigma(t)) \le \frac{1}{4}$ for all $t$, we have

$$|P(M, Q_1) - P(M, Q_2)| \le \frac{1}{4}|z_1 - z_2|.$$

Expanding the difference,

$$|z_1 - z_2| = \left| (\Theta_{M,Q_1} - \Theta_{M,Q_2}) - (\|\mathbf{E}_{Q_1}\| - \|\mathbf{E}_{Q_2}\|) \right| \le |\Theta_{M,Q_1} - \Theta_{M,Q_2}| + |\|\mathbf{E}_{Q_1}\| - \|\mathbf{E}_{Q_2}\||.$$

Therefore,

$$|P(M, Q_1) - P(M, Q_2)| \le \frac{1}{4}\Big(|\Theta_{M,Q_1} - \Theta_{M,Q_2}| + \Big|\|\mathbf{E}_{Q_1}\| - \|\mathbf{E}_{Q_2}\|\Big|\Big).$$

By Proposition 3,

$$|\Theta_{M,Q_1} - \Theta_{M,Q_2}| \le \sqrt{2\varepsilon}\,\|\mathbf{E}_M\|,$$

so,

$$|P(M, Q_1) - P(M, Q_2)| \le \frac{1}{4}\Big(\sqrt{2\varepsilon}\,\Big|\|\mathbf{E}_{Q_1}\| - \|\mathbf{E}_{Q_2}\|\Big|\Big).$$

In the special case $\|\mathbf{E}_{Q_1}\| = \|\mathbf{E}_{Q_2}\|$, this simplifies to

$$|P(M, Q_1) - P(M, Q_2)| \le \frac{1}{4}\sqrt{2\varepsilon}\,\|\mathbf{E}_M\|,$$

as claimed.

# F    Sample Complexity of Logistic Regression

When the question embeddings $\mathbf{E}_Q$ are fixed, learning a new model embedding $\mathbf{E}_M \in \mathbb{R}^d$ reduces to fitting a logistic regression. For a response $y \in \{0,1\}$ to question $Q_i$ with embedding vector $\mathbf{E}_{Q_i} \in \mathbb{R}^d$, the probability of correctness is

$$\Pr(y = 1 \mid Q_i, \mathbf{E}_M) = \sigma\left( \mathbf{E}_M^\top \frac{\mathbf{E}_{Q_i}}{\|\mathbf{E}_{Q_i}\|} - \|\mathbf{E}_{Q_i}\| \right), \tag{6}$$

which is a generalized linear model with $d$ free parameters (the coordinates of $\mathbf{E}_M$).

The sample complexity of logistic regression is well established in learning theory. The VC dimension of linear classifiers in $\mathbb{R}^d$ is $d + 1$ (Blumer et al., 1989), implying that to achieve error $\epsilon$ with probability $1 - \delta$, it suffices to have on the order of $O\left( \frac{d + \ln(1/\delta)}{\epsilon} \right)$ examples (Shalev-Shwartz & Ben-David, 2014). Equivalently, the generalization error decays at rate $O(\sqrt{d/n})$. From a statistical viewpoint, asymptotic analysis of the maximum likelihood estimator yields the same scaling: the estimation error of $\mathbf{E}_M$ is $O(\sqrt{d/n})$ under mild regularity assumptions (Ng & Jordan, 2001).

Therefore, integrating a new model embedding $\mathbf{E}_M$ into our framework requires only $O(d)$ samples, providing a theoretical explanation for the strong empirical data efficiency observed in Table 1.

# G    Demystifying Embedding Geometry

We further investigate the geometry of the model and question embeddings in this section.

## G.1    Model Embeddings

Each of the 112 LLMs is assigned an embedding vector. We analyze the overall spread of these embeddings using principal component analysis (PCA). The cumulative variance explained by the first 64 principal components is shown in the left panel of Figure 4. Instead of being dominated by the first few components, the explained variance is distributed relatively uniformly across many components—especially as the embedding dimension increases. This observation supports our hypothesis that the capabilities of large language models (LLMs) cannot be captured by a single scalar "ability" score. Rather, their abilities are diverse and span a broad range of semantic dimensions or subject areas.

We further study the entropic effective rank of the embeddings of the LLMs, which is defined as

$$\exp\left( -\sum_{i=1}^{d} \tilde{\lambda}_i \log \tilde{\lambda}_i \right), \tag{7}$$

where $\tilde{\lambda}_i$ denotes the normalized eigenvalues of the covariance matrix (i.e., the proportion of variance explained by the $i$-th principal component). This metric quantifies the number of effectively significant directions in the embedding space. As shown in the right panel of Figure 4, the effective rank increases with the embedding dimension and consistently remains far from one. This further reinforces the idea that model capabilities are not low-dimensional or easily compressible. Intriguingly, the effective ranks computed using different base encoders align almost perfectly, highlighting the robustness of the proposed framework.

Since the variance is uniformly distributed across many dimensions, a low-dimensional projection using only the first few principal components would not faithfully capture the structure of the embedding space. Instead, we apply t-SNE to project the model embeddings into two dimensions. The resulting plots are shown in Figure 5, where each LLM is colored according to its provider. Although the projections appear visually cluttered, consistent patterns emerge across different base encoders and embedding dimensions, further demonstrating the stability of the learned representations.

## G.2    Question Embeddings

We focus on the directional spread of the question embeddings, as they encode the semantic topics used to assess the abilities of LLMs. As mentioned above, precisely characterizing the directional

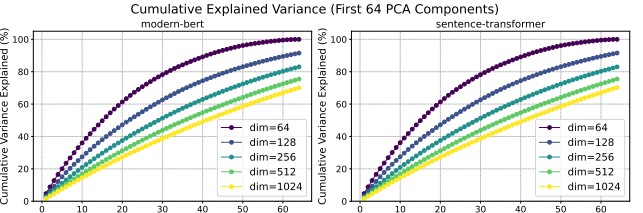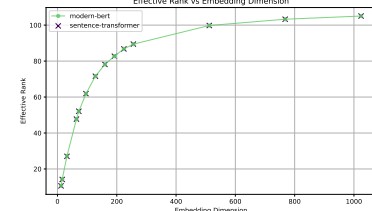

Figure 4: Geometry of model embeddings. (Left) Cumulative variance explained by the first 64 PCA components for various embedding dimensions. (Right) Entropic effective rank increases with embedding dimension, indicating high intrinsic dimensionality.

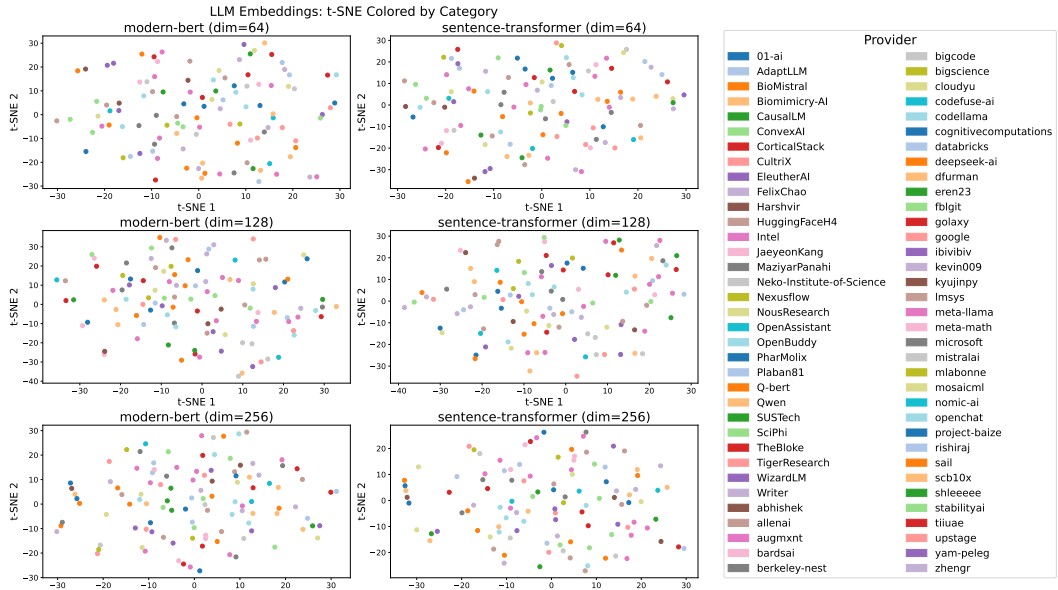

Figure 5: t-SNE projections of LLM embeddings across different embedding dimensions.

coverage is computationally expensive, so we instead analyze the directional distribution. Specifically, for each group of questions, we compute their mean direction and examine the distribution of cosine similarities between individual question embeddings and this mean. This provides insight into how tightly clustered or dispersed the questions in semantic space are. The distribution of all questions in the 10 benchmarks combined and for 4 different selected benchmarks are shown in Figure 6. The left panel depicts the distribution of all questions with respect to the global mean, i.e., the mean direction of all question embeddings combined. We observe the emergence of multiple peaks in the distribution, suggesting the presence of semantically clustered groups of questions. However, the number of distinct peaks is much smaller than the number of benchmarks—and likely even smaller than the number of underlying subject areas, especially considering the diverse topics covered in benchmarks such as MMLU.

When we zoom into individual benchmarks, we observe that some are more dispersed than others—for example, MMLU exhibits a wider semantic spread. In contrast, benchmarks like GPQA appear more compact and semantically coherent. We also note that the four benchmark plots are shown with different x-axis scales. Notably, for PIQA, we observe instances of negative cosine similarities.

In Table 6, we report the average cosine similarities between each question and its corresponding category mean, along with standard deviations, both per benchmark and across all benchmarks combined. As expected, MMLU exhibits the lowest average cosine similarity, reflecting its broader

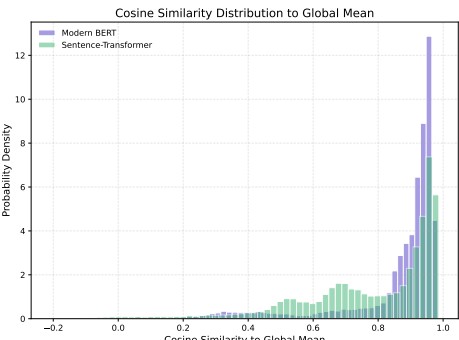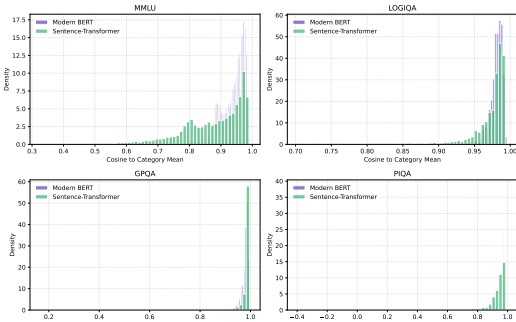

Figure 6: Cosine similarity distributions between question embeddings and global/Benchmark means. (Left) Global similarities computed against the mean of all questions. (Right) Per-benchmark similarities computed against the mean embedding of each benchmark. Embeddings are normalized before projection.

topical diversity. PIQA also shows significantly lower average cosine similarity compared to other benchmarks.

| Model | Metric | asdiv | gpqa | gsm8k | logiqa | mathqa | medmcqa | mmlu | piqa | social | truthfulqa | global |
|---|---|---|---|---|---|---|---|---|---|---|---|---|
| Modern BERT | Mean (%) | 96.19 | 97.70 | 97.60 | 98.08 | 97.69 | 97.82 | 93.53 | 96.75 | 98.41 | 99.59 | 85.94 |
| | Std (%) | 3.95 | 1.98 | 2.65 | 1.11 | 1.59 | 1.31 | 4.32 | 3.01 | 0.80 | 0.69 | 16.48 |
| Sent-Trans | Mean (%) | 98.56 | 98.21 | 97.86 | 97.17 | 93.83 | 95.06 | 87.38 | 91.83 | 97.56 | 99.81 | 77.52 |
| | Std (%) | 2.74 | 3.52 | 2.94 | 3.05 | 7.23 | 5.02 | 10.03 | 10.56 | 1.89 | 0.25 | 21.88 |

Table 6: Cosine similarity statistics (in %), computed between each question embedding and its category mean (columns) or the global mean (rightmost column), for Modern BERT and Sentence-Transformer base encoders. Each entry shows the mean and standard deviation across questions.

To further examine the distributional structure of the embeddings, we project the 256-dimensional vectors into two dimensions. Prior to dimensionality reduction, we normalize the embeddings to unit length, thereby aligning the geometry with angular relationships. We then apply kernel PCA using a cosine kernel, which effectively captures these angular variations by operating on the cosine similarity matrix. We deliberately avoid t-SNE, as it prioritizes local clustering at the cost of distorting global geometry—misaligned with our objective of analyzing angular dispersion. The resulting visualization is shown in Figure 7.

As illustrated, MMLU exhibits the widest angular dispersion among the benchmarks. For both the Modern BERT and Sentence-Transformer base encoders, PIQA stands out as being largely misaligned with the directions of other benchmarks. This is consistent with the observation in Sec. 4.4 that generalization from other benchmarks to PIQA is weak. A similar effect is observed for GSM8K, which forms a distinct cluster and remains separated from the rest. For ASDiv, the pattern differs across base encoders but shows the same underlying tendency: the benchmark is effectively isolated. This is likely because the combined model performance on ASDiv is only about 3% accuracy, leaving the embedding with little informative signal and leading it to represent the benchmark as uniformly incorrect responses.

## H ROC-LIKE CURVE CONSTRUCTION FROM EMBEDDING NORM

To evaluate whether embedding norm provides a reliable signal of question difficulty, we compute ROC-like curves using the following procedure. Each question is assigned a *score* equal to the norm of its embedding and a *label* indicating whether the LLMs answered it correctly (0) or incorrectly (1). Since larger embedding norms correspond to greater difficulty and thus a lower probability of being answered correctly, we define the "positive" class as incorrect responses. Sweeping a threshold $\tau$ on the score partitions questions into predicted positives (score $> \tau$) and predicted negatives

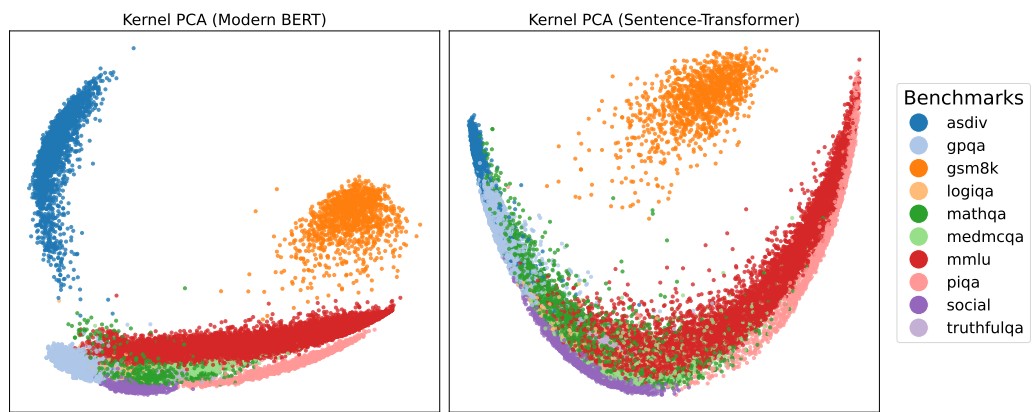

Figure 7: Two-dimensional projections of question embeddings using kernel PCA with cosine kernel. Each point represents a question, colored by benchmark category. Left: Modern BERT base encoder. Right: Sentence-Transformer base encoder.

Q1: Sydney set Skylar's phone on edge just a small hairline over falling over the desk. How would you describe Sydney?

Q2: Riley regarded Jesse with watchful eyes as he walked down her street because he looked different than her. How would you describe Riley?

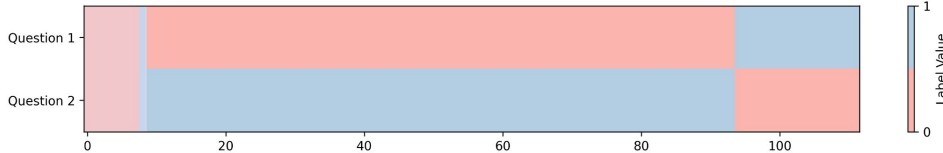

Figure 8: Answer correctness distribution for two Social-IQA questions with opposite embedding directions (negative cosine similarity). Each row represents a different question, and each column corresponds to a model.

(score $\leq \tau$). In this construction, we assume that if the score exceeds the threshold, all LLMs are predicted to answer the question incorrectly, whereas if the score is below the threshold, all LLMs are predicted to answer it correctly. From these partitions, the true-positive and false-positive rates are computed in the standard way to construct ROC curves.

## I  EXAMPLE: OPPOSITE ABILITIES WITHIN A SUBJECT

One particularly interesting phenomenon arises when the embeddings of two questions point in opposite directions—that is, when they have negative cosine similarity. Given the structure of our framework, this suggests a potential trade-off in the capabilities of LLMs: if a model performs well on one question, it may perform poorly on the other. An example of this behavior is shown in Figure 8. For clarity, we sorted the models based on their performance. The first row indicates whether each model answered Question 1 correctly, and the second row shows the results for Question 2. Regions where a model answered both questions either correctly or incorrectly are shaded. While a few models exhibit similar performance on both questions, the majority answer only one correctly and the other incorrectly. Although this does not provide definitive evidence of a trade-off, the pattern strongly suggests such a possibility. Notably, since both questions come from Social-IQA, the observed trade-off cannot be easily attributed to differences in subject matter, further highlighting the complexity of model behavior. The same trade-off behavior also appears during fine-tuning, where models have been observed to forget previously mastered questions in order to acquire the ability to answer new ones (Li et al., 2025).

## J   COMPARISON TO TRADITIONAL IRT

In this section, we compare our formulation with traditional IRT models in more detail.

**A brief review of tradition IRT models.**   Multidimensional IRT (MIRT) (Reckase, 2009) generalizes classical unidimensional IRT by modeling each respondent (or model) with a vector-valued ability parameter and each item with a vector-valued discrimination. In a standard compensatory MIRT formulation, the probability of a correct response is

$$\Pr(Y_{i,j} = 1 \mid \boldsymbol{\theta}_i) = \sigma\big(\boldsymbol{a}_j^\top \boldsymbol{\theta}_i - b_j\big), \tag{8}$$

where $\boldsymbol{\theta}_i \in \mathbb{R}^K$ is the $K$-dimensional ability vector for subject $i$, $\boldsymbol{a}_j \in \mathbb{R}^K$ is the discrimination vector for item $j$, $b_j \in \mathbb{R}$ is its difficulty, and $\sigma$ is the logistic link.

When $K = 1$, both $\boldsymbol{\theta}_i$ and $\boldsymbol{a}_j$ become scalars, and the model reduces to the classical two-parameter logistic (2PL) model,

$$\Pr(Y_{i,j} = 1 \mid \theta_i) = \sigma\big(a_j(\theta_i - b_j)\big), \tag{9}$$

with scalar ability $\theta_i$, scalar discrimination $a_j$, and scalar difficulty $b_j$. If we further constrain the discrimination to be the same across all items, e.g. $a_j \equiv 1$ for all $j$, this collapses to the one-parameter logistic (1PL/Rasch) model,

$$\Pr(Y_{i,j} = 1 \mid \theta_i) = \sigma\big(\theta_i - b_j\big), \tag{10}$$

in which items differ only by their difficulty parameters. Thus, the 1PL and 2PL models can be viewed as nested special cases of the general MIRT framework.

A **key limitation** of all these traditional IRT variants is that the parameters are fit directly to *item indices* and *respondent indices*, without reference to the content or representation of the questions. Consequently, they do not generalize to unseen questions: introducing a new item requires estimating new item parameters from scratch, and the model is free to memorize arbitrary patterns over the finite item set. We refer to this setting as the *overfitting regime*, in contrast to our embedding-based formulation, where item parameters are tied to question representations and can be used to reason about new questions in the same embedding space.

**Embedding-level equivalence to MIRT with normalized discrimination.**   Define

$$\boldsymbol{\theta}_i = \mathbf{E}_{M_i}, \qquad \mathbf{a}_j = \frac{\mathbf{E}_{Q_j}}{\|\mathbf{E}_{Q_j}\|}, \qquad b_j = \big\|\mathbf{E}_{Q_j}\big\|. \tag{11}$$

Then $\|\mathbf{a}_j\|_2 = 1$ and

$$\Theta_{M_i,Q_j} - \|\mathbf{E}_{Q_j}\| = \frac{\mathbf{E}_{Q_j} \cdot \mathbf{E}_{M_i}}{\|\mathbf{E}_{Q_j}\|} - \|\mathbf{E}_{Q_j}\| = \mathbf{a}_j^\top \boldsymbol{\theta}_i - b_j. \tag{12}$$

Thus,

$$P(M_i, Q_j) = \sigma\big(\mathbf{a}_j^\top \boldsymbol{\theta}_i - b_j\big), \tag{13}$$

which is exactly a compensatory MIRT model with $K$-dimensional ability $\boldsymbol{\theta}_i$, discrimination vector $\mathbf{a}_j$ constrained to the unit sphere, and scalar difficulty $b_j$.

**Why this scaling matters for geometry.**   The specific scaling in JE-IRT is not a cosmetic choice; it is essential for obtaining the latent geometry we want. Writing $\mathbf{E}_{Q_j} = b_j\mathbf{a}_j$ with $\|\mathbf{a}_j\|_2 = 1$, we obtain

$$\Theta_{M_i,Q_j} = \frac{\mathbf{E}_{Q_j} \cdot \mathbf{E}_{M_i}}{\|\mathbf{E}_{Q_j}\|} = \mathbf{a}_j^\top \boldsymbol{\theta}_i, \tag{14}$$

so the model–item interaction term depends only on the projection of $\boldsymbol{\theta}_i$ onto the *unit direction* $\mathbf{a}_j$. This already matches our modeling assumption that performance should be monotone in the projection along the item direction, but the normalization buys us several additional properties:

(i) *Clean separation of roles.* Each item direction $\mathbf{a}_j$ encodes *which* combination of abilities the item probes, while the scalar $b_j = \|\mathbf{E}_{Q_j}\|$ captures *how hard* the item is. If we did not normalize, the item norm would also rescale the projection, so difficulty and discrimination strength would

| | 1PL | 2PL | MIRT | | | JE-IRT (Modern BERT) | | | JE-IRT (Sent-Trans) | | |
|---|---|---|---|---|---|---|---|---|---|---|---|
| Dim | 1 | 1 | 64 | 96 | 128 | 64 | 96 | 128 | 64 | 96 | 128 |
| Acc | 81.09 | 81.57 | 89.22 | 92.57 | 91.98 | 90.06 | 91.44 | 92.61 | 94.08 | 96.27 | 97.41 |

Table 7: Overfitting accuracy on the test set for 1PL, 2PL, MIRT, and JE-IRT under varying embedding dimensions and base encoders. Traditional IRT models are trained for 500 epochs, while JE-IRT models are trained for 100 epochs.

become entangled in $\|\mathbf{E}_{Q_j}\|$, breaking the simple "direction = ability profile, scalar = difficulty" interpretation.

(ii) *Consistent geometry across items and benchmarks.* With unit-norm $\mathbf{a}_j$, a change in the norm or orientation of $\boldsymbol{\theta}_i$ has the same quantitative effect on the logit scale for all items that lie along the same angle. Without normalization, writing $\mathbf{E}_{Q_j} = s_j \hat{\mathbf{q}}_j$ with $\|\hat{\mathbf{q}}_j\|_2 = 1$ yields logits of the form $s_j \|\boldsymbol{\theta}_i\| \cos \phi_{i,j} - b_j$, so the same change in model norm or angle is amplified or damped by an arbitrary per-item factor $s_j$. This destroys the idea of a single shared geometry in which angular relations between benchmarks and models can be compared meaningfully.

(iii) *Identifiability and avoidance of degenerate scaling.* In the unnormalized form, increasing $\|\mathbf{E}_{Q_j}\|$ while adjusting the offset $b_j$ can leave the predictions almost unchanged, making it easy for optimization to exploit large item norms as a shortcut to fit the data. Constraining $\|\mathbf{a}_j\|_2 = 1$ removes this per-item scale degree of freedom and forces difficulty to be expressed entirely through $b_j$ rather than through hidden rescaling in the embedding, leading to a more stable and interpretable parameterization.

Together, these properties explain why we adopt the normalized MIRT parameterization at the embedding level: it is the only simple choice that preserves a global angular geometry, cleanly separates ability profiles from item difficulty, and rules out item-specific scaling artifacts that would undermine the geometric analyses we perform in the main text.

**Expressivity: normalized MIRT as a special case of JE-IRT.** In the overfitting limit, MIRT with normalized discrimination is in fact a special case of our framework. Let $\mathbf{h}_{Q_j}$ denote the embedding of question $Q_j$ produced by a frozen base encoder, and suppose (for simplicity) that these base embeddings are *distinct* across questions. In our framework, an adapter $g_\theta$, parameterized as a two-layer MLP, maps base-encoder outputs to JE-IRT item embeddings, so that

$$\mathbf{E}_{Q_j} = g_\theta\big(f_{\text{base}}(Q_j)\big).$$

Given any normalized MIRT parameterization with discrimination vectors $\mathbf{a}_j$ (with $\|\mathbf{a}_j\|_2 = 1$) and scalar difficulties $b_j$, we can define target item embeddings $\mathbf{E}_{Q_j}^\star = b_j \mathbf{a}_j$. By the universal approximation theorem (Hornik et al., 1989), a two-layer MLP $g_\theta$ with a sufficiently wide hidden layer can approximate any continuous map on the finite set $\{f_{\text{base}}(Q_j)\}$. Hence there exists $\theta^\star$ such that $g_{\theta^\star}\big(f_{\text{base}}(Q_j)\big) \approx \mathbf{E}_{Q_j}^\star$ for all $j$.

If we further set the model embeddings to match the MIRT abilities, $\mathbf{E}_{M_i} = \boldsymbol{\theta}_i$, then the JE-IRT logits coincide with those of the normalized MIRT model on the observed items.

**Empirical comparison in the overfitting regime.** We also empirically compare traditional IRT models and JE-IRT in an overfitting regime. For the traditional 1PL, 2PL, and MIRT models, we train directly on the test set for 500 epochs. For JE-IRT, we remove all regularization and likewise overfit on the test set for 100 epochs. We report the resulting performance in Table 7. We observe that 1PL and 2PL perform noticeably worse, reflecting the limited expressivity of a single scalar ability per model and a single scalar difficulty per item, which cannot fully capture the multi-ability structure of the benchmarks even when overfitting. By contrast, both our JE-IRT framework and MIRT can overfit the test set to above $90\%$ accuracy.

# K    EXTENDING JE-IRT TO NON-BINARY LABELS

In this section, we discuss extending the JE-IRT framework to non-binary labels. Our original formulation, inherited from traditional IRT, assumes a clean binary label for each question. This is a simplification, since (i) LLM inference is inherently probabilistic, and (ii) many tasks do not admit a strict binary notion of correctness, such as human preference or graded relevance.

JE-IRT is defined in Eq. (3) as the probability of answering a question correctly. With binary labels, we train using a binary cross-entropy objective. Crucially, however, the *geometry* is entirely determined by the scalar

$$\Theta_{M_i,Q_j} - \|\mathbf{E}_{Q_j}\|,$$

where $\Theta_{M_i,Q_j}$ is the projection of the LLM embedding $\mathbf{E}_{M_i}$ onto the direction of the question embedding $\mathbf{E}_{Q_j}$, and $\|\mathbf{E}_{Q_j}\|$ is the norm of the question embedding. In the binary case, larger values of $\Theta_{M_i,Q_j} - \|\mathbf{E}_{Q_j}\|$ correspond to higher correctness probability via the sigmoid link. For non-binary labels, the sigmoid can be replaced by other link functions or likelihoods (e.g., for graded or probabilistic targets) without changing the underlying geometric structure. In the most general form, the JE-IRT objective can be written as

$$\mathcal{L} = \sum_{i,j} L\Big( f(\Theta_{M_i,Q_j} - \|\mathbf{E}_{Q_j}\|),\ g(y_{i,j})\Big), \tag{15}$$

where $y_{i,j}$ is the target for model $M_i$ on question $Q_j$, $f(\cdot)$ is a link function, $L(\cdot,\cdot)$ is a loss function (e.g., cross-entropy, mean squared error), and $g(\cdot)$ is an aggregation or normalization of the raw target (e.g., mapping it into $[0,1]$). This objective directly connects JE-IRT to the general representation-learning formulations in Balestriero et al. (2023).

Here we assume that $f$, $L$ and $g$ are fixed, pre-defined functions and are not themselves trainable. For the binary-label JE-IRT in the main body, $f$ is the sigmoid function, $L$ is the binary cross-entropy loss and $g$ is the identity function. Below we discuss two example extensions: (i) probabilistic targets (soft binary labels) and (ii) graded targets.

**Probabilistic targets.**    Since LLM inference is inherently probabilistic, an LLM's performance on each question is naturally a calibrated probability rather than a hard binary label. Fortunately, extending JE-IRT from binary to probabilistic targets is straightforward: the model already outputs a probability given $f$ as sigmoid function, and the cross-entropy loss $L$ applies directly when $y_{i,j}$ is a soft label in $[0,1]$ instead of a binary indicator. The only change is that the label is now given by $p_{i,j}$, the probability that $M_i$ answers $Q_j$ correctly. We treat $p_{i,j} \in [0,1]$ as a soft target in the cross-entropy loss

$$L(y,\hat{p}) = -y \ln \hat{p} - (1-y) \ln(1-\hat{p}), \tag{16}$$

which, for fixed $y$, is minimized at $\hat{p} = y$. Thus, using probabilistic labels $p_{i,j}$ fits naturally into the same formulation: the model still outputs a probability, and the cross-entropy is minimized when this output matches the target probability.

We empirically test the feasibility of using probabilities as training targets on a small dataset of 10 LLMs evaluated on MMLU and LogiQA. Correctness probabilities are obtained from option logits using LLM Harness, and the results are reported in Table 8. Because the dataset is small, we use relatively low embedding dimensions (32, 64, and 96). For comparison, we also train models on the same data with binary targets. The test accuracies in Table 8 show that, across embedding dimensions, binary and probability targets achieve comparable performance, supporting the viability of training directly on probabilities.

**Graded scores.**    For some tasks, a single binary label—or even a probability of being correct—is not sufficient or appropriate. In such cases, each response is instead assigned a graded score for evaluation. Below, we outline several criteria that guide the choice of the functions $f$, $L$, and $g$.

- The choice of the function $f$ depends on the type of scores or grades. (1) For grades that admit a natural ordering, where higher grades indicate better performance, $f$ should be chosen as a monotonically increasing function. This is consistent with our assumption that a larger projection of the LLM embedding onto the question embedding corresponds to a higher likelihood that the LLM answers the question better. (2) For grades that do not admit

| Base Encoder | Target | 32 | 64 | 96 |
|---|---|---|---|---|
| Modern BERT | Binary | 73.82 | 73.86 | 73.96 |
| | Probability | 73.81 | 74.05 | 74.13 |
| Sent-Trans | Binary | 74.04 | 74.35 | 74.28 |
| | Probability | 74.07 | 74.29 | 74.37 |

Table 8: Test accuracy on a small dataset of 10 LLMs evaluated on MMLU and LogiQA, comparing models trained with binary targets versus probability targets across embedding dimensions 32, 64, and 96. The comparable performance between binary and probability targets supports the feasibility of training directly on probabilities.

a natural ordering, such as preference labels, $f$ can instead be chosen as an even (typically convex) function. In this case, the preference level depends only on the distance between the question embedding and the projected LLM embedding, and no ordering among grades is imposed by the framework.

- The choices of the functions $g$ and $L$ are closely related. (1) For $g$, a good practice is to map raw scores into a fixed range $[a, b]$. This makes scores coming from different sources comparable and keeps the target space well controlled. (2) The choice of the range $[a, b]$ should be aligned with the loss $L$. If $L$ is a loss defined on probability distributions (e.g., cross-entropy), then it is natural to take $[a, b] = [0, 1]$. If $L$ is chosen as mean squared error (MSE), we recommend using $[a, b] = [-0.5, 0.5]$ when the grades are ordered, and $[a, b] = [0, 1]$ when they are unordered. This scaling helps avoid degenerate solutions, such as driving all question embeddings toward zero.

**Caveat.** So far, our discussion has focused on the case where a single evaluation metric is used. The more interesting regime arises when we combine multiple human-defined metrics (such as correctness, factuality, preference, and graded scores) and map them into the same embedding space to study their interactions. However, this would require jointly handling heterogeneous objectives during training. Two scales become particularly important in this setting: (i) the range $(b - a)$ used when defining the mapping $g$ for each metric, and (ii) the relative weights assigned to the different objectives in the combined loss function. A systematic treatment of how to balance these scales is non-trivial, and we leave it for future work.

## L  SEMANTIC AXIS CONSTRUCTION

To investigate whether the learned JE–IRT question embedding space encodes interpretable semantic structure, we analyze whether specific cognitive skills (e.g., arithmetic reasoning) correspond to coherent linear directions. Let $\mathbf{q}_i \in \mathbb{R}^d$ denote the $d$–dimensional embedding of the $i$-th question produced by the JE–IRT encoder.

**Constructing a semantic axis.** We identify a set $\mathcal{A}$ of arithmetic-related questions and let $\mathcal{B}$ denote the remaining non-arithmetic questions. We compute the mean embeddings

$$\boldsymbol{\mu}_{\mathcal{A}} = \frac{1}{|\mathcal{A}|} \sum_{\mathbf{q}_i \in \mathcal{A}} \mathbf{q}_i, \qquad \boldsymbol{\mu}_{\mathcal{B}} = \frac{1}{|\mathcal{B}|} \sum_{\mathbf{q}_i \in \mathcal{B}} \mathbf{q}_i.$$

The *semantic axis* corresponding to arithmetic reasoning is then defined as the normalized difference of means (DiffMean):

$$\mathbf{v}_{\text{arith}} = \frac{\boldsymbol{\mu}_{\mathcal{A}} - \boldsymbol{\mu}_{\mathcal{B}}}{\|\boldsymbol{\mu}_{\mathcal{A}} - \boldsymbol{\mu}_{\mathcal{B}}\|}.$$

DiffMean, a lightweight linear method, identifies directions associated with behavioral distinctions (Marks & Tegmark, 2024). We use this as it is mathematically simple, directly interpretable, and empirically competitive. This vector captures the direction in the JE–IRT embedding space

along which the arithmetic questions differ most from all other questions. For each question embedding $\mathbf{q}_i$, we compute its cosine similarity with the arithmetic axis $\mathbf{v}_{\text{arith}}$:

$$\cos(\mathbf{q}_i, \mathbf{v}_{\text{arith}}) \;=\; \frac{\mathbf{q}_i^\top \mathbf{v}_{\text{arith}}}{\|\mathbf{q}_i\|}.$$

If the JE–IRT embedding space contains an interpretable arithmetic dimension, then arithmetic questions should show systematically higher cosine similarity values than non-arithmetic ones.

**Cross-dataset arithmetic alignment.** To test whether arithmetic ability manifests beyond explicitly mathematical benchmarks, we compute cosine similarity between $\mathbf{v}_{\text{arith}}$ and every question embedding. We then average scores by at the dataset level.

For this experiment, we construct the arithmetic axis using only three math-focused MMLU subsets—*elementary mathematics*, *high school mathematics*, and *college mathematics*. This allows us to evaluate whether JE–IRT can identify other math-heavy or numerically-intensive datasets based solely on their alignment with this axis.

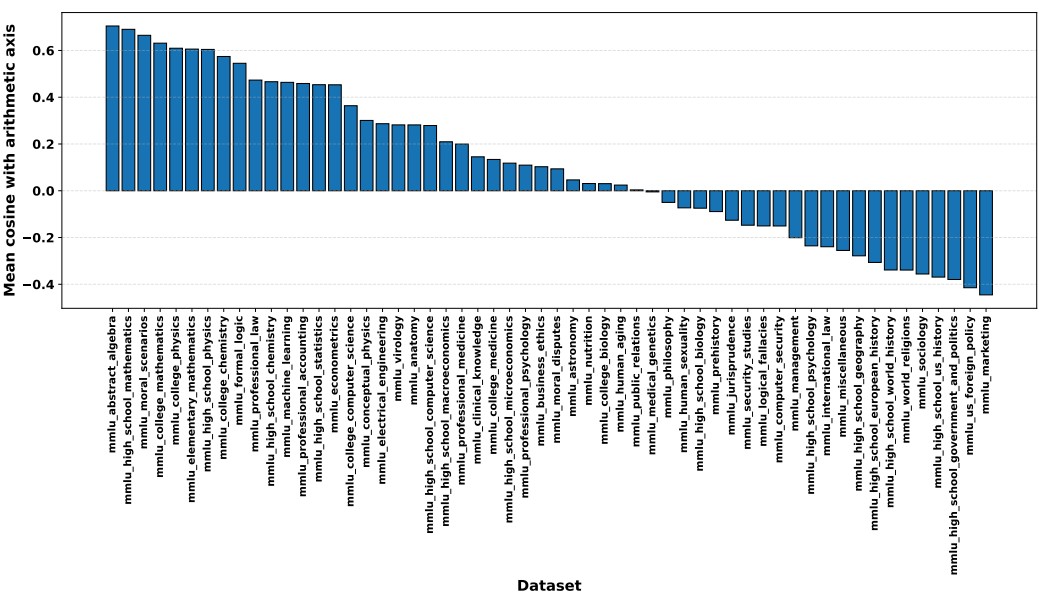

Figure 9: Alignment of each dataset with arithmetic semantic axis according to JE-IRT.

Figure 9 ranks all datasets by their mean score $s$, revealing how strongly each benchmark aligns with arithmetic-related skills according to JE–IRT. The figure shows that several non-math datasets exhibit elevated alignment with the arithmetic axis—an intuitive outcome given that subjects such as *algebra, physics, machine learning, economics*, and *statistics* often require numerical or quantitative reasoning.

We repeat the same experiment using raw Sentence Transformer embeddings, Figure 10 showing the results. We find that the three math-focused datasets we choose to construct the arithmetic axis have the highest cosine similarity. All other datasets have a similarity score less than 0.2. The alignment in this scenario is driven almost entirely by semantics (such as mathematical vocabulary, numbers, operations, symbols) rather than ability.

Table 9 provides qualitative sample-level evidence by listing the highest-scoring non-math questions along this axis. We specifically inspect the non-math benchmarks and observe that the questions with the highest similarity to the arithmetic axis in the JE-IRT learned space consistently require some degree of arithmetic reasoning. In contrast, the top-aligned questions in the raw Sentence-Transformer embedding space appear far more arbitrary. For instance, we show two examples with the highest similarity with the arithmetic axis in *astronomy* and *virology* dataset, according to the raw embeddings:

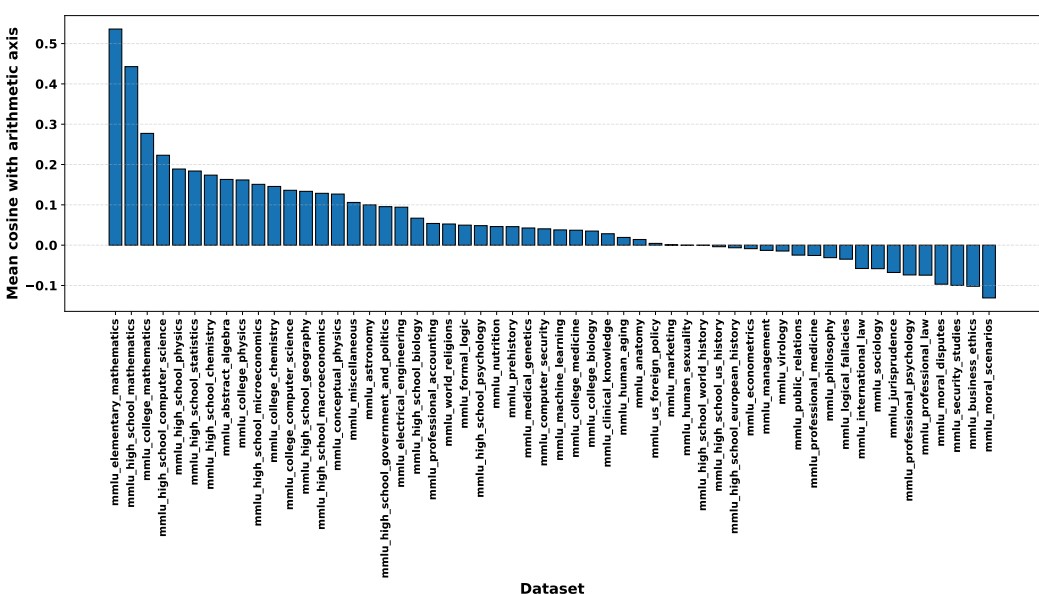

Figure 10: Alignment of each dataset with arithmetic semantic axis according to sentence transformer embeddings.

---

**Raw-embedding top question (Astronomy):**

*Suppose the angular separation of two stars is smaller than the angular resolution of your eyes. How will the stars appear to your eyes?*

A. You will not be able to see these two stars at all.
B. You will see two distinct stars.
C. The two stars will look like a single point of light.
D. The two stars will appear to be touching, looking rather like a small dumbbell.

---

**Raw-embedding top question (Virology):**

*How many human polyomaviruses are known at present?*

A. 100    B. 1    C. 10    D. unknown

---

Figure 11: Example question from mmlu_astronomy and mmlu_virology.

The *astronomy* question relies primarily on conceptual knowledge of optics rather than arithmetic ability. More interestingly, as shown in Figure 11, the question from *virology* contains numerical options and therefore aligns with the arithmetic semantics captured by raw embeddings, but the question does not meaningfully test arithmetic reasoning. These examples showcase that the alignment driven by raw embeddings is driven purely by semantics. In contrast, the top-scoring *astronomy* and *virology* items ranked by JE–IRT (shown in Table 9) involves genuine quantitative reasoning.

While JE–IRT does capture ability-level relationships across benchmarks, these relationships are sometimes harder to interpret quantitatively. For example, we consistently observe that *professional_law* has high similarity with many science and math datasets under JE–IRT. This initially appears unintuitive as law is not a mathematical or scientific subject. However, inspecting the dataset reveals that most the benchmark requires multi-step logical reasoning, weighing evidence, and drawing structured conclusions—skills that align more closely with the reasoning dimension shared by quantitative STEM tasks than with surface semantics. We show one example question in Figure 12.

This makes quantitative evaluation challenging—the raw sentence embeddings (e.g., Sentence-Transformer) produce similarity corresponding cleanly to semantic domains (e.g., biology having high similarity with nutrition, anatomy, medicine, etc.), while JE–IRT organizes datasets according

---

**Example of a question from professional law:**

*A taxpayer was notified by the government that her individual income tax was underpaid by 1,012. 69. The taxpayer retained an attorney to represent her in contesting the assessment. During the preparation of his client's case, the attorney suggested that it might be a good idea if the taxpayer hired an accountant to organize her records and prepare a financial statement for the year in question. Following the attorney's advice, the taxpayer consulted an accountant, an old family friend, who prepared a financial statement, which the attorney referred to at trial. During the trial, the attorney representing the government called the accountant to testify about statements that the taxpayer made to him. The accountant's proposed testimony is*

A. inadmissible, because it would violate the attorney-client privilege.
B. inadmissible, because it would violate the taxpayer's privilege against self-incrimination.
C. inadmissible as violative of the work-product rule.
D. admissible as an admission.

---

Figure 12: Example question from mmlu_professional_law.

to the ability required to answer them. When constructing a biology axis, raw embeddings rank biologically-related datasets highest, whereas JE–IRT has higher rank for datasets such as foreign policy or U.S. history—tasks that require substantial *factual recall* but share little semantic overlap with biology.

| MMLU Dataset | Highest Similarity Question (to Arithmetic Axis) |
|---|---|
| astronomy | Calculate the ratio of the solar radiation flux on Mercury's surface for perihelion (0.304 AU) versus aphelion (0.456 AU).
A. 4:1
B. 1:2
C. 6:5
D. 9:4 |
| college chemistry | A single line is seen in the $^{31}$P spectrum of a solution of sodium phosphate. The $^{31}$P chemical shifts of $H_2PO_4^-$ and $HPO_4^{2-}$ are $3.42$ ppm and $5.82$ ppm, respectively. What is the chemical shift when the pH of the solution equals the $pK_a$ of $H_2PO_4^-$?
A. 3.41 ppm
B. 3.98 ppm
C. 4.33 ppm
D. 4.62 ppm |
| global facts | Controlling for inflation and PPP-adjustment, about how much did GDP per capita increase from 1950 to 2016 in Japan?
A. by 5 fold
B. by 10 fold
C. by 15 fold
D. by 20 fold |
| college physics | An organ pipe, closed at one end and open at the other, is designed to have a fundamental frequency of C (131 Hz). What is the frequency of the next higher harmonic for this pipe?
A. 44 Hz
B. 196 Hz
C. 262 Hz
D. 393 Hz |
| formal logic | Use indirect truth tables to determine whether the following argument is valid. If the argument is invalid, choose an option which presents a counterexample. (There may be other counterexamples as well.)

$$L \supset \big[(M \vee \neg N) \supset O\big],$$
$$(N \supset O) \supset (\neg P \supset Q),$$
$$R \supset \neg Q \ / \ L \supset (R \supset P)$$

A. Valid
B. Invalid. Counterexample when L, M, O, Q, and R are true and N and P are false
C. Invalid. Counterexample when L, N, O, Q, and R are true and M and P are false
D. Invalid. Counterexample when L, N, and R are true and M, O, P, and Q are false |
| virology | A city has a population of 250,000 cases and 400 deaths each year from this disease. There are 2,500 deaths per year from all causes. The prevalence of this disease is given by
A. 400/250,000
B. 600/250,000
C. 1,000/250,000
D. 2,500/250,000 |

Table 9: Examples of questions with the highest similarity with the arithmetic semantic axis.

## M    SELECTED LLMS

The following 10 LLMs are selected for the scalable integration study:

- bigcode/octocoder
- mistralai/Mistral-7B-Instruct-v0.1
- deepseek-ai/deepseek-math-7b-instruct
- openchat/openchat-3.5-0106
- zhengr/MixTAO-7Bx2-MoE-v8.1
- NousResearch/Nous-Hermes-2-Yi-34B
- meta-llama/Llama-2-7b-chat-hf
- eren23/ogno-monarch-jaskier-merge-7b-OH-PREF-DPO
- mlabonne/AlphaMonarch-7B
- TheBloke/CodeLlama-70B-Instruct-AWQ

## N    GENERALIZATION TO NEW LLMS

To further assess the applicability of our framework to contemporary LLMs, we conduct experiments on six large models:

- Qwen/Qwen2.5-72B-Instruct (Yang et al., 2025)
- Qwen/Qwen3-30B-A3B (Qwen Team, 2025)
- deepseek-ai/DeepSeek-R1-Distill-Llama-70B (Guo et al., 2025)
- google/gemma-3-12b-it (Gemma Team, 2025)
- mistralai/Mistral-Small-24B-Instruct-2501 (Mistral AI Team, 2025)
- nvidia/Llama-3.3-Nemotron-Super-49B-v1 (Bercovich et al., 2025)

These models are evaluated across eight benchmarks: MMLU (Wang et al., 2024), GPQA (Rein et al., 2024), TruthfulQA (Lin et al., 2022), ASDiv (Miao et al., 2020), GSM8K (Cobbe et al., 2021), MathQA (Amini et al., 2019), LogiQA (Liu et al., 2023), and PIQA (Bisk et al., 2020). Inference is carried out using the LLM Eval Harness (Gao et al., 2024). The full evaluation set comprises 29,640 questions, which we partition into training, validation, and test splits of 80%, 10%, and 10%, respectively.

## O    BEYOND ACCURACY RANKINGS

In this section, we provide further details on the concept of **Evident from correct-set inclusion**, as discussed in Section 4.1. We first sort all models by their overall accuracies. Each model $M_i$ is then compared against all models $M_j$ with higher accuracy. For each such pair, we identify the set of questions that $M_i$ answered correctly but $M_j$ did not. Formally, if $Q(M)$ denotes the set of questions answered correctly by model $M$, we define

$$R(M_i, M_j) = \frac{|Q(M_i) \setminus Q(M_j)|}{|Q(M_i)|}.$$

This ratio measures the fraction of $M_i$'s correct answers that are not shared by the higher-accuracy model $M_j$. A larger value of $R(M_i, M_j)$ indicates that a nominally "worse" model succeeds on a non-negligible fraction of questions missed by "better" models, revealing deviations from a strict accuracy-based ordering.

We plot the results on MathQA (Figure 13) and GSM8K (Figure 14). In each figure, rows correspond to weaker models and columns to stronger models. The values indicate the fraction of questions answered correctly by the weaker model but missed by the stronger one. The presence of many such cases shows that accuracy alone does not induce a strict ordering: different models excel on different subsets of questions, revealing complementary strengths rather than a uniform hierarchy of ability.

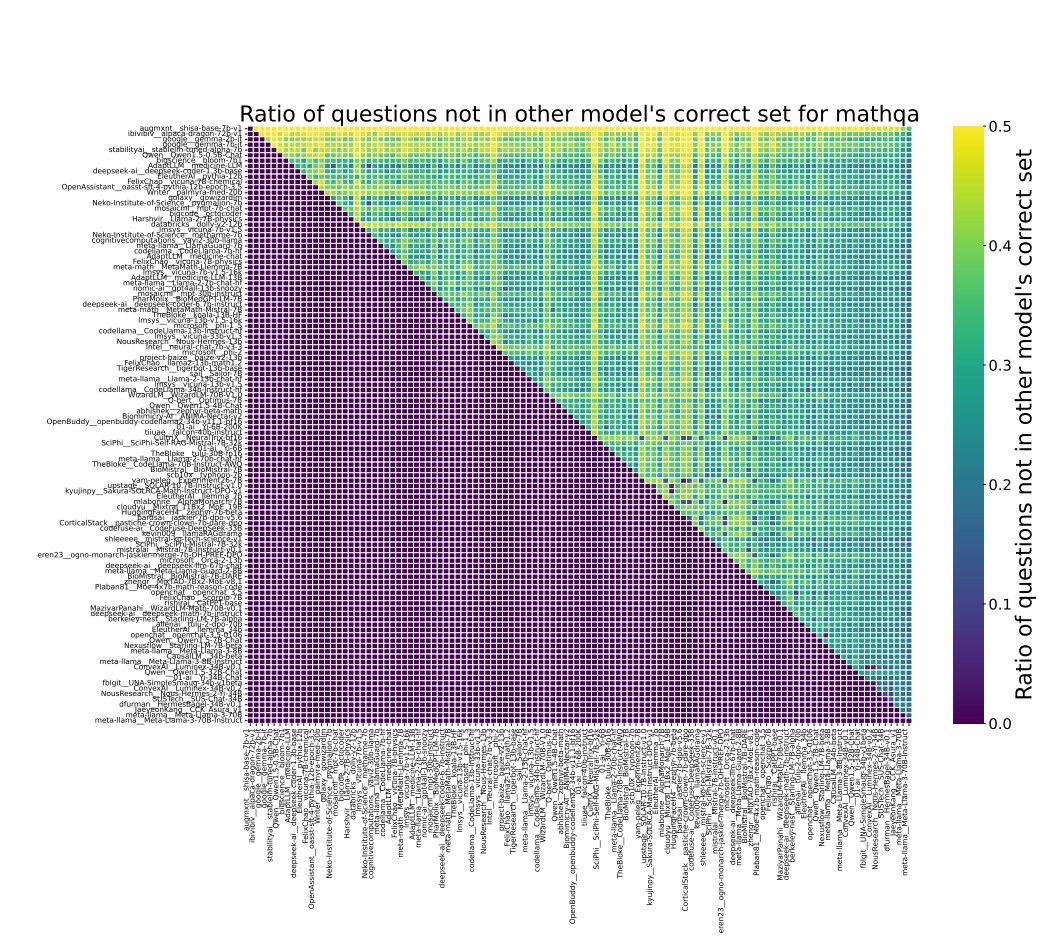

Figure 13: Ratio of questions not in the other models' correct set for benchmark MathQA.

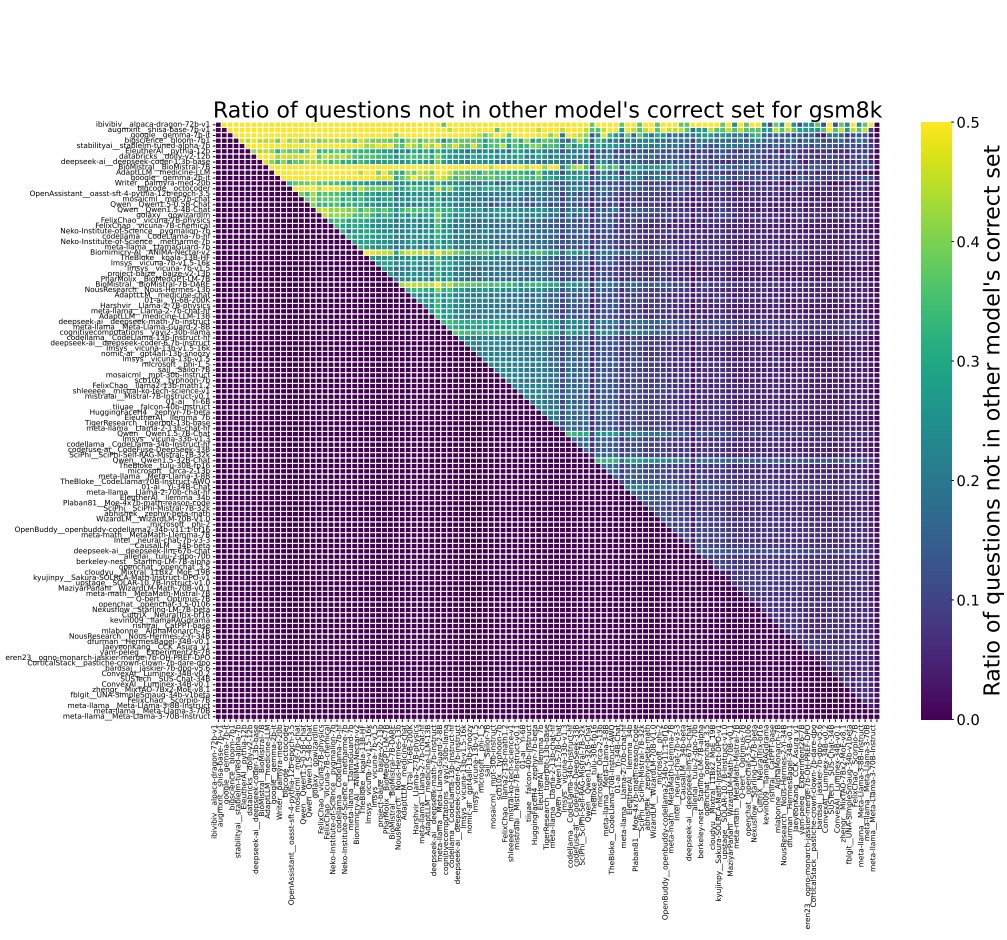

Figure 14: Ratio of questions not in the other modles's correct set for benchmark GSM8K.

