# OpenReview forum: "A Geometric Lens on LLM Abilities through Joint Embedding Item Response Theory"
_ICLR.cc/2026/Conference — Submitted to ICLR 2026_

### Official Review · Reviewer_FPY1 · 2025-11-01

**Soundness:** 3
**Presentation:** 3
**Contribution:** 3
**Rating:** 4
**Confidence:** 2

**Summary:**

The paper proposes JE-IRT, a joint embedding variant of item response theory that maps both large language models and benchmark questions into a shared geometric space where direction encodes topical alignment and norm captures difficulty. Trained on the EmbedLLM correctness dataset across 112 models and 10 benchmarks, the method predicts item-level correctness via the projection of model embeddings onto question directions, achieving better accuracy than scalar-ability baselines while remaining parameter-efficient. Extensive analyses show that the learned geometry reflects interpretable structure, supports adding new LLMs via lightweight fine-tuning, and reveals that model-induced subject clusters only partially match human-defined categories.

**Strengths:**

The paper grounds its framework in clear theoretical motivations, demonstrating via empirical evidence that traditional 2PL IRT assumptions fail for contemporary LLM evaluations. JE-IRT produces competitive accuracy with lower-dimensional embeddings, and its inductive bias yields consistent gains across models and benchmarks, suggesting broad applicability. The authors validate interpretability claims by correlating embedding norms with question difficulty, assessing directional alignment for OOD behavior, and analyzing clustering structure, providing richer insight than aggregate leaderboard scores. Scalability experiments that onboard both held-out and newer LLMs using only a single embedding underscore the practicality of the approach.

**Weaknesses:**

The evaluation hinges entirely on the EmbedLLM dataset and a small set of base encoders, leaving open whether the learned geometry transfers to other evaluation suites or task formats beyond multiple-choice correctness. While the framework is positioned as interpretable, the paper provides limited qualitative case studies illustrating how practitioners would act on the embeddings, and the taxonomy analysis stops short of human validation. Computational costs for training and updating the embeddings are not reported, making it hard to judge feasibility for organizations with many models or rapidly evolving benchmarks. The method also assumes clean correctness labels and does not explore robustness to annotation noise or adversarial inputs.

**Questions:**

How sensitive is JE-IRT to the choice of frozen base encoder, and could domain-specific encoders or instruction-tuned representations further improve semantic alignment without retraining question embeddings? When adding new models, does the single-embedding fine-tuning ever require regularization to prevent overfitting to scarce observations, especially for highly specialized systems? Can the authors share qualitative examples where the learned geometry surfaces actionable routing or evaluation insights that human-designed subject tags would miss? Finally, how would the framework accommodate non-binary evaluation signals such as partial credit, calibrated probabilities, or human preference judgments?

---

> ### Author Response · Authors · 2025-11-20
> **Responses to Reviewer Comments Part 1**
>
> We appreciate the reviewer’s insightful comments and the opportunity to clarify our contributions. We address each question and concern in the responses that follow. We have also added three new sections to the appendix: Appendix J (Comparison to Traditional IRT), Appendix K (Extending JE-IRT to Non-Binary Labels), and Appendix L (Semantic Axis Construction) to provide additional clarification on these aspects.
>
> --------
>
> >The evaluation hinges entirely on the EmbedLLM dataset and a small set of base encoders, leaving open whether the learned geometry transfers to other evaluation suites or task formats beyond multiple-choice correctness.
>
> >Finally, how would the framework accommodate non-binary evaluation signals such as partial credit, calibrated probabilities, or human preference judgments?
>
> JE-IRT is not tied to a particular evaluation suite, base encoder, or task format. In our main
> experiments we use EmbedLLM and binary correctness labels simply because they provide a
> large, unified collection of benchmarks and models, and binary signals are the most widely
> available and comparable across datasets. The framework itself only requires a supervision
> signal for each (model, question) pair, and Section 4.4 shows that the learned geometry
> transfers in a predictable way to held-out benchmarks (e.g., PIQA, MMLU, GSM8K), with
> generalization behavior controlled by directional alignment in the embedding space.
>
> More generally, Appendix K formulates JE-IRT beyond binary labels via
>
> $
> \mathcal{L}=
> \sum_{i,j}
> L\Bigl(
>   f(\Theta_{M_i,Q_j} - \lVert E_{Q_j}\rVert),
>   g(y_{i,j})
> \Bigr),
> $
>
> where $y_{i,j}$ is an arbitrary target (e.g., partial credit, probability, or human preference
> score), $f$ is a link function, $L$ a loss, and $g$ a normalization of the raw signal. We then
> theoretically discuss two concrete non-binary extensions: (1) probabilistic outputs and (2) graded
> or continuous scores. We also empirically demonstrate, using a pilot experiment with 10 LLMs
> evaluated on MMLU and LogiQA, that replacing binary labels with logits-derived probabilities
> yields stable training and test performance comparable to the binary variant across multiple
> embedding dimensions. This provides empirical evidence that JE-IRT naturally extends to
> non-binary supervision without altering the underlying joint embedding geometry.
>
> ### Binary vs. Probability Targets
>
> | Base Encoder  | Target      | 32    | 64    | 96    |
> |---------------|-------------|-------|-------|-------|
> | Modern BERT   | Binary      | 73.82 | 73.86 | 73.96 |
> | Modern BERT   | Probability | 73.81 | 74.05 | 74.13 |
> | Sent-Trans    | Binary      | 74.04 | 74.35 | 74.28 |
> | Sent-Trans    | Probability | 74.07 | 74.29 | 74.37 |
>
> *Table: Test accuracy on a small dataset of 10 LLMs evaluated on MMLU and LogiQA, comparing models trained with binary targets versus probability targets across embedding dimensions 32, 64, and 96. The comparable performance between binary and probability targets supports the feasibility of training directly on probabilities (see Appendix K for details).*
>
>
> -------
>
> >The method also assumes clean correctness labels and does not explore robustness to annotation noise or adversarial inputs.
>
> JE-IRT, like most IRT-style models, assumes clean correctness labels. Robustness to annotation noise or adversarial inputs is an important but orthogonal direction that we do not explore in this work and regard as a natural target for future extensions.
>
>
> ------
>
> >Computational costs for training and updating the embeddings are not reported, making it hard to judge feasibility for organizations with many models or rapidly evolving benchmarks.
>
> JE-IRT is computationally lightweight in our setting. During the research process, we first generate text embeddings for all ~34k questions in EmbedLLM; using Modern BERT and SentenceTransformer encoders, this takes under 5 minutes on a single RTX 4090. Training JE-IRT on the full EmbedLLM matrix with embedding dimension up to 1024 also fits on a single RTX 4090 and completes in under 2 hour. With the question-embedding branch frozen, adding a new LLM only requires learning its model embedding $E_M$, which takes less than 20 minutes on a single RTX 4090. We will add these computational details in the final version.

---

> ### Author Response · Authors · 2025-11-20
> **Responses to Reviewer Comments Part 2**
>
> >While the framework is positioned as interpretable, the paper provides limited qualitative case studies illustrating how practitioners would act on the embeddings, and the taxonomy analysis stops short of human validation.
>
> JE-IRT is interpretable primarily in a geometric sense: questions and models inhabit a shared low-dimensional space in which direction encodes required abilities and norm encodes difficulty. Section 4.6 analyzes this structure using propositions, subject-coverage metrics, and cross-subject overlap patterns. To illustrate how practitioners could act on these embeddings, we expanded our qualitative analysis in the revision.
>
> In Appendix L, we construct an arithmetic semantic axis via a DiffMean contrast and find out questions with high cosine similarities with the axis. Inspecting high-similarity items reveals that, beyond explicitly tagged math problems, JE-IRT highlights physics, chemistry, and global-facts questions that primarily test quantitative or symbolic reasoning (Table 9), showing how practitioners could identify ability-intensive subsets that cut across subject labels. We also show that JE-IRT recovers broader ability-level relationships across subjects: for example, professional law appears similar to several science and math datasets (Figure 12) not because of shared vocabulary, but because many law questions require multi-step reasoning and structured inference. Such qualitative case studies demonstrate how the learned geometry can guide practitioners in analyzing ability profiles or constructing skill-targeted sub-benchmarks.
>
> By contrast, applying the same semantic-axis construction to frozen base encoder embeddings yields directions dominated by superficial lexical cues (e.g., numeric tokens) and fails to surface non-math questions requiring genuine quantitative reasoning (Appendix L, Figure 11). This suggests that JE-IRT introduces additional ability-sensitive structure beyond the encoder alone. While a full human evaluation of the taxonomy is an important direction for future work, our focus here is on establishing the geometric interpretability and empirical utility of the framework.
>
> -------
>
> >How sensitive is JE-IRT to the choice of frozen base encoder, and could domain-specific encoders or instruction-tuned representations further improve semantic alignment without retraining question embeddings?
>
> In our experiments, JE-IRT does show some dependence on the choice of frozen base encoder, but our main qualitative conclusions are robust across encoders. In the early stages of this work we tried four encoders (SentenceTransformer, Longformer, ModernBERT, and Qwen-3 embeddings) and chose to report the two strongest ones for space and clarity. Interestingly, the Qwen-3 encoder, although very strong on standard embedding benchmarks, was less effective in our setting and reached about 72% test accuracy. This suggests that JE-IRT benefits more from representations that preserve factors correlated with item “difficulty” and “ability profile” rather than generic semantic similarity. The two encoders we report (ModernBERT and SentenceTransformer) yield consistent trends on orientation vs. norm, OOD behavior, and clustering, even though their absolute accuracies differ.
>
> Domain-specific or instruction-tuned encoders are fully compatible with JE-IRT and are a promising avenue to further improve semantic alignment in specialized domains. In our framework, one can simply regenerate question embeddings with a new encoder and refit the JE-IRT adapter and model embeddings, which is computationally cheap. We did not fine-tune encoders or train domain-specific ones in this work due to data and compute constraints, but designing encoders explicitly optimized for JE-IRT’s objectives (e.g., better preserving difficulty-related factors) is a natural and appealing direction for future work.

---

> ### Author Response · Authors · 2025-11-20
> **Responses to Reviewer Comments Part 3**
>
> >When adding new models, does the single-embedding fine-tuning ever require regularization to prevent overfitting to scarce observations, especially for highly specialized systems?
>
> In our experiments on adding new models (Section 4.3), we use only light weight decay and no additional regularization. Because the question-embedding branch is frozen, fitting a new model reduces to learning a single vector $E_M$ via logistic regression (Appendix F), with the embedding dimension already selected based on performance over ~100 base models. Under this setup we did not observe overfitting, even when training on a small fraction of the available questions, and held-out accuracy remained stable.
>
> That said, more aggressive regularization (e.g., stronger weight decay or priors that tie new model embeddings toward a reference family) would be straightforward to add in settings with extremely scarce data or highly specialized systems. We leave a systematic study of such regimes to future work.
>
> ---------
>
> >Can the authors share qualitative examples where the learned geometry surfaces actionable routing or evaluation insights that human-designed subject tags would miss?
>
> Although routing is a natural downstream use case of our framework, our work focuses on the shared geometry itself and does not develop a specific routing policy. Rather, we view JE-IRT as providing a shared geometric representation that routing or evaluation methods could build on. In particular, improved item-level prediction and a low-dimensional ability/difficulty structure could, in principle, inform routing decisions, although practical routing would also need to account for additional considerations (e.g., cost, latency, model availability) that are beyond the scope of this paper.
>
> Regarding qualitative insights beyond human subject tags, Section 4.6 quantitatively analyzes their overlap with JE-IRT clusters, and Appendix L further examines semantic directions derived from the learned geometry. For example, an “arithmetic” axis constructed from model directions reveals many questions originally tagged under non-math subjects whose content primarily requires arithmetic or quantitative reasoning; Appendix L (Table 9) provides concrete examples. We see these cases as illustrating how the learned geometry provides ability-based structure that human-designed subject tags may not fully capture.

---

### Official Review · Reviewer_25eA · 2025-11-02

**Soundness:** 2
**Presentation:** 3
**Contribution:** 3
**Rating:** 6
**Confidence:** 2

**Summary:**

This paper proposes a geometric item-response framework, JE-IRT, to evaluate the multi-dimensional ability of LLMs by incorporating semantics and difficulties of questions.

**Strengths:**

* Presents a geometric IRT framework, JE-IRT, embedding both models and questions into a shared space, with their dot product determining the probability of a correct response. Provides interpretable question vector attributes with direction capturing semantics and norm capturing difficulty.
* Clear motivation showing a total order of LLM performance using scalar abilities does not hold, and the need to move to multidimensional interactions.
* Scalable to evaluating new LLMs through only fine-tuning their embedding instead of full retraining.
* Interpretable geometric validation showing directional alignment predicts out-of-distribution drops and norms relate to difficulty. Clustering the embeddings reveals subject divisions that are different from human-defined categories.
* JE-IRT outperforms baselines in the prediction accuracy of LLMs on benchmarks.
* The methodology is clearly presented and well written with descriptive figures.

**Weaknesses:**

* Compare novelty and prediction performance with MIRT baseline: JE-IRT is similar to Multidimensional-IRT, an existing IRT framework (Multidimensional item response theory models, Mark D Reckase, 2009). MIRT also includes a dot product between student (model) and item (question) vectors. While the authors provide a geometric interpretation to this formulation, the novelty compared to MIRT should be clearly explained. Further, recent applications of IRT in NLP already use embeddings of question content and model representations (see the IRT-Router paper, which the authors cite).
* Compare against other IRT models on predictive performance. Since the authors extend IRT to JE-IRT, the performance of existing IRT models like 1-PL, 2-PL, and MIRT should be compared against.
* Design choices of JE-IRT should be clarified with intuition and theoretical justification. For ex, why divide by the norm of the question embedding to get model abilities (eq 2), why use the norm of the question embedding as difficulty, etc.
* The authors motivate why a scalar-based total ordering of model abilities doesn’t hold. Similarly, using the norm of the question embedding to get scalar-abilities would lead to a total ordering of question difficulties. I assume there exist harder questions that can be solved by weaker models and vice versa. Can the authors verify this? If yes, shouldn’t a similar multidimensional interaction between model vectors and question vectors be used to obtain question difficulties? Is this already done in MIRT to get difficulty (I’m not sure)?

**Questions:**

* The authors cluster the question embeddings and show that it is not well aligned with human-defined subjects. The authors state that items that cluster together require similar abilities from the LLM. I wasn’t able to find the justification for this claim.
* The paper’s main goal is LLM evaluation. The authors cite Fluid Language Model Benchmarking, a recent work using IRT-based LLM evaluation, which focuses on efficiency and includes evaluation metrics such as efficiency (small number of questions for LLM evaluation), validity (generalization to other benchmarks), variance, etc. While the authors show that new LLMs can be added efficiently, is JE-IRT sample efficient? Can the Fisher information idea from Fluid Language Model Benchmarking be applied to JE-IRT? Apart from LLM response correctness prediction, can results on recovering the LLM ordering at a per-benchmark level be shown?
* Generalization to some new datasets (piqa, mmlu, gsm8k) shows significant drops in performance. How can this be mitigated? What fraction of questions from these datasets would reduce the drop? For example, adding 10% of Q from gsm8k leads to similar performance as in-distribution.
* Since the L2 norm is not differentiable everywhere, the authors should clarify how backprop is performed during training (sub-gradients, etc).

---

> ### Author Response · Authors · 2025-11-20
> **Responses to Reviewer Comments Part 1**
>
> We appreciate the reviewer’s thoughtful and detailed critique. We address all questions and concerns in the following responses and provide further evidence supporting our claims. We have also added three new sections to the appendix: Appendix J (Comparison to Traditional IRT), Appendix K (Extending JE-IRT to Non-Binary Labels), and Appendix L (Semantic Axis Construction) to provide additional clarification on these aspects.
>
> -----
>
> >Compare novelty and prediction performance with MIRT baseline: JE-IRT is similar to Multidimensional-IRT, an existing IRT framework (Multidimensional item response theory models, Mark D. Reckase, 2009). MIRT also includes a dot product between student (model) and item (question) vectors. While the authors provide a geometric interpretation to this formulation, the novelty compared to MIRT should be clearly explained.
>
> >Compare against other IRT models on predictive performance. Since the authors extend IRT to JE-IRT, the performance of existing IRT models like 1-PL, 2-PL, and MIRT should be compared against.
>
> Our framework is inspired by IRT, but its goal is different: JE-IRT studies the interaction between questions and LLMs through a shared, geometrically interpretable embedding space. Question embeddings encode both the ability profile they test (via direction) and their difficulty (via norm), which allows us to analyze model–question relations in a way that classical IRT formulations do not.
>
> Traditional IRT and MIRT, by contrast, learn item parameters (difficulty or discrimination) as a lookup table tied to the correctness of each question, without using question content. As a result, they do not generalize to unseen questions. JE-IRT instead ties question embeddings to content through a shared encoder, so once trained it can be applied to new questions without refitting.
>
> Because classical MIRT is undefined on unseen items, a fair comparison is only possible when training and testing on the same data. Appendix J empirically shows that 1PL and 2PL perform markedly worse in this regime, while MIRT and JE-IRT both reach high accuracy (above 90%), with JE-IRT matching or exceeding MIRT’s expressivity. Appendix J also theoretically shows that compensatory MIRT is a special case of JE-IRT when the adapter is a sufficiently wide two-layer MLP.
>
> ### Overfitting Accuracy Comparison
>
> | Model (Dim) | 1PL (1) | 2PL (1) | MIRT (64) | MIRT (96) | MIRT (128) | JE-IRT BERT (64) | JE-IRT BERT (96) | JE-IRT BERT (128) | JE-IRT Sent-Trans (64) | JE-IRT Sent-Trans (96) | JE-IRT Sent-Trans (128) |
> |-----|-----|-----|------------|------------|-------------|--------------------|--------------------|---------------------|---------------------------|---------------------------|----------------------------|
> | Acc | 81.09 | 81.57 | 89.22 | 92.57 | 91.98 | 90.06 | 91.44 | 92.61 | 94.08 | 96.27 | 97.41 |
>
> *Table: Overfitting accuracy on the test set for 1PL, 2PL, MIRT, and JE-IRT under varying embedding dimensions and base encoders. Traditional IRT models are trained for 500 epochs, while JE-IRT models are trained for 100 epochs.*
>
> -------
>
> >Further, recent applications of IRT in NLP already use embeddings of question content and model representations (see the IRT-Router paper, which the authors cite).
>
> While both JE-IRT and IRT-Router use IRT-style components with embeddings, their goals and parameterizations differ.
>
> IRT-Router is designed as a *routing* architecture. For each item $i$ and model $M$, it learns a vector-valued discrimination parameter $\mathbf{a}_i$, a vector-valued model representation $\boldsymbol{\theta}_M$, and a scalar difficulty $b_i$, which are then used inside a cost-aware routing objective with additional transformation layers. These quantities are optimized to support routing decisions under budget constraints, not to define a stable, shared latent geometry. As a result, the parameters playing the role of “ability”, “discrimination”, and “difficulty” are closely tied to routing-specific losses and heads, and are not meant to be directly compared across benchmarks or used for geometric analysis.
>
> JE-IRT, in contrast, is explicitly formulated as a joint embedding model for *analysis*. Each model and each question is represented by a single vector in a common space, and the same logits that drive prediction are exactly the objects we study. There are no extra routing heads or budget-aware transformations that reshape the latent variables. This geometry-first design yields a clean latent space in which (i) directions encode ability profiles that can be compared across models and benchmarks, and (ii) norms/offsets capture difficulty.
>
> The angular analysis and ability–difficulty comparisons we perform in JE-IRT rely on this structured embedding formulation and do not transfer directly to the more entangled parameterization used by IRT-Router.

---

> ### Author Response · Authors · 2025-11-20
> **Responses to Reviewer Comments Part 2**
>
> >Design choices of JE-IRT should be clarified with intuition and theoretical justification. For ex, why divide by the norm of the question embedding to get model abilities (eq 2), why use the norm of the question embedding as difficulty, etc.
>
> In this work we adopt the inductive bias that *direction encodes an ability profile and norm encodes difficulty*. We write each question embedding as
> $u_Q = E_Q / \|E_Q\|$ (an ability profile over skills such as arithmetic, commonsense, coding, etc.) and $\|E_Q\|$ (difficulty along that profile). The model’s ability on $Q$ is then given by the projection
> $\theta_{M,Q} = u_Q \cdot E_M$, and we define the success probability as:
>
> $
> P(M,Q) = \sigma\bigl(\theta_{M,Q} - \|E_Q\|\bigr).
> $
>
> The benefit of this formulation is that it makes the geometry directly interpretable: questions with different directions $u_Q$ correspond to different ability profiles, and questions with larger $\|E_Q\|$ correspond to harder items along the same ability profile. This choice is consistent with classical IRT: when the embedding dimension is one, it naturally recovers the standard 2PL model, while in higher dimensions it extends that inductive bias to a multidimensional, geometrically interpretable setting.
>
> Finally, dividing by the question norm isolates the ability profile. Since
>
> $
> \theta_{M,Q}
> = \frac{E_Q \cdot E_M}{\|E_Q\|}
> = u_Q \cdot E_M,
> \qquad
> u_Q = \frac{E_Q}{\|E_Q\|},
> $
>
> $\theta_{M,Q}$ depends only on $u_Q$ and is invariant to $\|E_Q\|$. Thus $\theta_{M,Q}$ measures how much of the relevant skill the model has, while $\|E_Q\|$ acts as a difficulty threshold along that skill. The underlying ability does not change just because we ask an easier or harder question in the same semantic direction.
>
>
>
>
>
>
> ------
>
>
> >The authors motivate why a scalar-based total ordering of model abilities doesn’t hold. Similarly, using the norm of the question embedding to get scalar-abilities would lead to a total ordering of question difficulties. I assume there exist harder questions that can be solved by weaker models and vice versa. Can the authors verify this? If yes, shouldn’t a similar multidimensional interaction between model vectors and question vectors be used to obtain question difficulties? Is this already done in MIRT to get difficulty (I’m not sure)?
>
> We appreciate this question, as it touches a subtle but important point about what “difficulty” means in a multidimensional setting.
>
> We examined two well-accepted definitions of difficulty: (i) the classical 2PL IRT difficulty (Section 4.1), and (ii) an empirical difficulty based on how many LLMs answer a question correctly (Appendix L). Under *both* notions, we observe the pattern the reviewer describes: some questions that appear “difficult” by either scalar measure are solved by weaker models, and some “easy” questions are missed by stronger models. This is one of the main motivations for JE-IRT: both “ability” and “difficulty” arise from high-dimensional model–question interactions and cannot be fully captured by a single global scalar.
>
> JE-IRT is explicitly multidimensional on *both* sides. Each model has an embedding $E_M \in \mathbb{R}^d$ and each question has an embedding $E_Q \in \mathbb{R}^d$. The logit depends on their interaction:
>
> $
> \Theta_{M,Q} - \|E_Q\|,
> $
>
> where $\Theta_{M,Q}$ is an angular term and $\|E_Q\|$ is a radial term. Geometrically, the *direction* of $E_Q$ encodes which combination of abilities the question tests, while its *length* acts as a difficulty *within that local direction*. As formalized in Proposition 2, $\|E_Q\|$ is only meaningfully comparable when two questions have directions within a small angular neighborhood. Outside such a cone, we see exactly the cross-over behavior the reviewer mentions: a model that is weak along one direction can still outperform a stronger model on questions whose embeddings point in a different direction.
>
> From this perspective, JE-IRT already treats both models and questions in a multidimensional way. In the new Appendix L on semantic axes (e.g., arithmetic vs. non-arithmetic), we project question embeddings onto interpretable ability directions and study how each question loads on these axes. In a qualitative study (Table 9), we find that questions from very different surface subjects can in fact rely mainly on arithmetic rather than on domain-specific knowledge.

---

> ### Author Response · Authors · 2025-11-20
> **Responses to Reviewer Comments Part 3**
>
> >The authors cluster the question embeddings and show that it is not well aligned with human-defined subjects. The authors state that items that cluster together require similar abilities from the LLM. I wasn’t able to find the justification for this claim.
>
> Our claim that “items that cluster together require similar abilities” follows directly from how
> JE-IRT is defined and validated. By design, the *direction* of a question embedding encodes
> which abilities it requires: we write $\Theta_{M,Q} = u_Q \cdot E_M$ with $u_Q = E_Q / \|E_Q\|$,
> so ability depends only on the direction $u_Q$ and not on the norm. This is further formalized in
> our theory (Propositions 2–3). Section 4.4 then provides empirical support that *ability coverage*
> correlates with *direction coverage* in the embedding space.
>
> Because of this, in our clustering analysis we explicitly *normalize* question embeddings and
> cluster them using cosine distance, so that clusters are determined purely by direction and therefore
> group together questions that require similar abilities under our model. Additionally, Appendix L
> studies semantic axes (e.g., an arithmetic direction) derived from these directions and shows that
> questions with high projections on a given axis share coherent skill requirements, providing further
> empirical support that directions in JE-IRT reflect underlying abilities. Finally, traditional
> subject-based benchmark groupings implicitly make an analogous assumption—that questions from
> the same subject require similar abilities—so the comparison between subject labels and JE-IRT
> clusters is precisely about whether human-defined subjects align with the ability structure inferred
> from LLM behavior.
>
>
>
> -------
>
>
>
> >The paper’s main goal is LLM evaluation. The authors cite Fluid Language Model Benchmarking, a recent work using IRT-based LLM evaluation, which focuses on efficiency and includes evaluation metrics such as efficiency (small number of questions for LLM evaluation), validity (generalization to other benchmarks), variance, etc. While the authors show that new LLMs can be added efficiently, is JE-IRT sample efficient? Can the Fisher information idea from Fluid Language Model Benchmarking be applied to JE-IRT? Apart from LLM response correctness prediction, can results on recovering the LLM ordering at a per-benchmark level be shown?
>
> We see JE-IRT as complementary to Fluid Language Model Benchmarking rather than a direct competitor. Our goal is to expose the *geometry* of model–question interactions, while Fluid focuses on *test efficiency* and benchmark design.
>
> JE-IRT is sample efficient once the question geometry is learned. As discussed in Appendix F, onboarding a new LLM reduces to logistic regression in $\mathbb{R}^d$, with sample complexity $O(d)$ for estimating the model embedding. Empirically, Tables 1 and 2 show that, with frozen question embeddings, using only $10$–$20\%$ of an LLM's responses recovers almost all of the joint-training performance, indicating good sample efficiency in practice.
>
> The Fisher-information ideas from Fluid can in principle be layered on top of JE-IRT. Our framework provides a smooth logit
>
> $
> \Theta_{M,Q} - \|E_Q\|,
> $
>
> so one can compute per-item gradients and Fisher information with respect to the model embedding and use them to select informative questions. We have not implemented such an active-testing policy here, but view this as promising future work.
>
> Finally, JE-IRT can recover conventional orderings when desired. Aggregating predicted correctness over all items in a benchmark yields a predicted accuracy per model, and ranking by this tracks the empirical benchmark ordering. However, our main focus is the underlying ability space and question–LLM interactions. Current benchmarks are organized using human priors about subject structure. They do not reflect how LLMs internally structure abilities. As a result, a single aggregate score is often not sufficient. JE-IRT is designed to reveal this richer geometry instead of collapsing it to one dimension.
>
> -----
>
> >Since the L2 norm is not differentiable everywhere, the authors should clarify how backprop is performed during training (sub-gradients, etc).
>
> PyTorch handles the L2 norm using its standard subgradient.
> The only concern is dividing by $\|E_q\|$ when it becomes zero; we add a small $\varepsilon$ to avoid
> this.
> In practice, question embeddings never collapse to zero because almost no items are solved by all
> LLMs.
> We did not observe any gradient instability. We will add this clarification in the final version.

---

> ### Author Response · Authors · 2025-11-20
> **Responses to Reviewer Comments Part 4**
>
> >Generalization to some new datasets (piqa, mmlu, gsm8k) shows significant drops in performance. How can this be mitigated? What fraction of questions from these datasets would reduce the drop? For example, adding 10% of Q from gsm8k leads to similar performance as in-distribution.
>
> To address the observed drops on unseen benchmarks, we conduct an adaptation experiment on
> GSM8K, PIQA, and MMLU where we augment the JE-IRT training set with a small fraction of
> questions from each target benchmark and retrain the adapter and model embeddings. As shown
> in the tables below, adding only 10% of questions from these benchmarks already yields a substantial
> recovery of performance. When the fraction reaches 50%, performance is very close to the 100%-data setting,
> indicating that a relatively small subset (around 10%–50%) is sufficient to significantly reduce
> the generalization gap.
>
> ### GSM8K
>
> | Base Encoder  | 0%    | 10%   | 30%   | 50%   | 100%  |
> |---------------|-------|-------|-------|-------|-------|
> | Modern BERT   | 59.05 | 73.54 | 73.80 | 74.55 | 74.86 |
> | Sent-Trans    | 59.05 | 69.27 | 73.11 | 73.93 | 73.94 |
>
>
> ### PIQA
>
> | Base Encoder  | 0%    | 10%   | 30%   | 50%   | 100%  |
> |---------------|-------|-------|-------|-------|-------|
> | Modern BERT   | 26.67 | 76.52 | 78.40 | 79.52 | 79.96 |
> | Sent-Trans    | 28.81 | 74.85 | 77.95 | 78.25 | 78.66 |
>
>
> ### MMLU
>
> | Base Encoder  | 0%    | 10%   | 30%   | 50%   | 100%  |
> |---------------|-------|-------|-------|-------|-------|
> | Modern BERT   | 46.50 | 67.20 | 67.86 | 67.98 | 68.40 |
> | Sent-Trans    | 50.49 | 65.61 | 66.59 | 67.13 | 67.84 |

---

### Official Review · Reviewer_NTra · 2025-11-03

**Soundness:** 3
**Presentation:** 3
**Contribution:** 3
**Rating:** 4
**Confidence:** 3

**Summary:**

This paper proposes JE-IRT (Joint Embedding Item Response Theory), a novel framework for evaluating large language models (LLMs) through a geometric generalization of item response theory (IRT). Instead of assigning each model and question a scalar ability and difficulty as in traditional IRT, JE-IRT embeds both in a shared vector space. In this space:

1. The direction of a question embedding captures its semantics (topic/specialization).
2. The norm encodes its difficulty.
3. The dot product projection between model and question embeddings determines the probability of a correct answer.

JE-IRT explains key empirical patterns observed in LLM evaluation: (i) no universal ordering of LLMs across tasks, and (ii) smooth variation of performance across related questions. The method is trained on the EmbedLLM correctness dataset (112 models, 10 benchmarks), and evaluated for scalability, interpretability, and generalization. The learned geometry reveals that embedding norms correlate with question difficulty, while directional alignment predicts out-of-distribution (OOD) generalization. Furthermore, clustering in the learned space shows that LLMs’ internal organization of topics partially diverges from human-defined subject boundaries.

**Strengths:**

1. The paper introduces a geometric formulation of IRT, where model–question interactions are represented through inner products in a shared space rather than scalar parameters. This is conceptually elegant and unifies item difficulty, semantic similarity, and model ability under a single geometric lens.
2. The theoretical formulation is well-motivated and supported by two formal propositions.

**Weaknesses:**

1. JE-IRT’s applicability to non-multiple-choice or free-form generation tasks remains unclear; the binary correctness assumption is acknowledged as a limitation but not empirically addressed.

**Questions:**

1. Have you explored whether specific semantic axes (e.g., reasoning, factual recall, arithmetic) emerge in the learned space? Visual examples could greatly enhance the interpretability claims.
2. Could JE-IRT be extended to handle graded or probabilistic outputs (e.g., continuous scores or multiple-choice logits)? Would replacing binary targets with model confidence distributions improve fit and interpretability?
3. Could the JE-IRT space support continual updates as new benchmarks or question types are added, without retraining the adapter network?
4. The cosine alignment between benchmark means correlates with OOD accuracy drops. Could this be formalized as a predictive diagnostic tool for estimating when a model will fail on a new dataset?

---

> ### Author Response · Authors · 2025-11-20
> **Responses to Reviewer Comments Part 1**
>
> We are grateful for the reviewer’s careful evaluation of our work and the valuable comments provided. We address each point in detail below, and we have also added three new sections to the appendix: Appendix J (Comparison to Traditional IRT), Appendix K (Extending JE-IRT to Non-Binary Labels), and Appendix L (Semantic Axis Construction) to provide additional clarification on these aspects.
>
> ------
>
> >JE-IRT’s applicability to non-multiple-choice or free-form generation tasks remains unclear; the binary correctness assumption is acknowledged as a limitation but not empirically addressed.
>
> >Could JE-IRT be extended to handle graded or probabilistic outputs (e.g., continuous scores or multiple-choice logits)?
>
>
> JE-IRT itself does not assume a multiple-choice format; it only requires a supervision signal for each model–question pair. In our main experiments this signal is instantiated as a binary correctness label, but the framework is agnostic to how that label is obtained (e.g., from exact match, reference-based metrics, or human judgment). Thus, free-form generation tasks are compatible with JE-IRT as soon as their outputs are mapped to correctness/quality scores.
>
> In the revised manuscript we make this explicit in Appendix K, where we extend JE-IRT beyond binary labels. We show that the training objective can be written in the general form
>
> $
> \mathcal{L}=
> \sum_{i,j}
> L\Bigl(
>   f(\Theta_{M_i,Q_j} - \lVert E_{Q_j}\rVert),
>   g(y_{i,j})
> \Bigr),
> $
>
> where $y_{i,j}$ is an arbitrary target for model $M_i$ on question $Q_j$, $f$ is a link function, $L$ is a loss, and $g$ is a (possibly non-linear) normalization of the raw target (e.g., mapping scores into $[0,1]$). The binary setting in the main text corresponds to $y_{i,j} \in \{0,1\}$, $f$ equal to the sigmoid, $L$ the binary cross-entropy, and $g$ the identity.
>
> Appendix K then theoretically discusses two concrete non-binary extensions: (1) probabilistic outputs and (2) graded scores. We also empirically demonstrate, using a pilot experiment with 10 LLMs evaluated on MMLU and LogiQA, that replacing binary labels with logits-derived probabilities yields stable training and test performance comparable to the binary variant across multiple embedding dimensions. This provides empirical evidence that JE-IRT naturally extends to non-binary supervision without altering the underlying joint embedding geometry.
>
>
> | Base Encoder  | Target       | 32    | 64    | 96    |
> |--------------|--------------|-------|-------|-------|
> | Modern BERT  | Binary       | 73.82 | 73.86 | 73.96 |
> | Modern BERT  | Probability  | 73.81 | 74.05 | 74.13 |
> | Sent-Trans   | Binary       | 74.04 | 74.35 | 74.28 |
> | Sent-Trans   | Probability  | 74.07 | 74.29 | 74.37 |
>
> *Table: Test accuracy on a small dataset of 10 LLMs evaluated on MMLU and LogiQA, comparing models trained with binary targets versus probability targets across embedding dimensions 32, 64, and 96. The comparable performance between binary and probability targets supports the feasibility of training directly on probabilities.*
>
> In summary, the binary instantiation in the main text is a modeling choice, not a limitation: Appendix K gives a general non-binary formulation and a proof of concept showing that JE-IRT extends to probabilistic and graded targets.
>
>
> ------
> >Would replacing binary targets with model confidence distributions improve fit and interpretability?
>
> Replacing binary targets with model confidence scores does not change the geometric interpretation
> of JE-IRT (direction = ability profile; norm = difficulty), but it can in principle provide a richer
> supervision signal on the logit scale. With soft targets $y \in [0,1]$ and cross-entropy,
>
> $
> \mathcal{L}(z,y)= -\big[y\log\sigma(z)+(1-y)\log(1-\sigma(z))\big],
> $
>
> the pointwise optimum satisfies $z^\star = \mathrm{logit}(y)$, i.e., the logit is pulled toward a finite
> target. In contrast, for hard labels $y \in \{0,1\}$ the optimum tends to $z^\star \to \pm\infty$ in the
> separable limit, which encourages extreme logits. Thus soft labels provide a more graded teaching
> signal while leaving the underlying joint-embedding geometry unchanged.
>
> As discussed in the previous answer and in Appendix K, we implement this extension by treating
> correctness probabilities $p_{i,j}$ (from model logit distributions) as soft labels in the same
> JE-IRT objective. A small pilot with 10 LLMs (Appendix K, Table 8) finds comparable performance
> to the binary setting; potential benefits in practice will depend on per-model calibration of
> confidences.

---

> ### Author Response · Authors · 2025-11-20
> **Responses to Reviewer Comments Part 2**
>
> >Could the JE-IRT space support continual updates as new benchmarks or question types are added, without retraining the adapter network?
>
> Yes. In our formulation, the base encoder and adapter together define a question embedding branch, and the learned JE-IRT space can be reused as new benchmarks arrive. If desired, one can continue training the same adapter on the expanded question set starting from its existing parameters (i.e., via continual fine-tuning rather than reinitialization).
>
> If both the base encoder and adapter are frozen, and the new benchmark probes skills that are largely absent from the training distribution, the frozen question branch may misplace its items.
>
> ------
>
> >Have you explored whether specific semantic axes (e.g., reasoning, factual recall, arithmetic) emerge in the learned space? Visual examples could greatly enhance the interpretability claims.
>
>
> To explore whether specific semantic axes emerge in the learned space, we added an analysis in Appendix L where we construct an arithmetic semantic axis in the JE-IRT question space using a DiffMean contrast between arithmetic and non-arithmetic questions. We then compute the cosine similarity between each question and this axis, and inspect the items with the highest similarities. Beyond explicitly tagged math problems, many high-similarity questions come from other domains (e.g., physics, chemistry, global facts) where solving the item still requires quantitative or symbolic reasoning (see Table 9 in Appendix L for examples). This suggests that the learned direction captures an underlying ability dimension (arithmetic/symbolic reasoning) rather than a narrow subject label or superficial lexical pattern, supporting our interpretability claims. While JE-IRT does capture ability-level relationships across benchmarks, these relationships can be hard to interpret in a purely quantitative way. For instance, professional law consistently appears similar to many science and math datasets under JE-IRT (see Figure 12 in Appendix L), not because it is a STEM subject, but because many questions require multi-step reasoning, weighing evidence, and drawing structured conclusions. These are the abilities that align with the reasoning dimension shared by quantitative tasks.
>
>
> By contrast, applying the same procedure directly to embeddings from the frozen base encoder yields an axis whose top-ranked items are dominated by questions with explicit numeric tokens or arithmetic keywords that do not necessarily require arithmetic ability (examples are shown in Figure 11 in Appendix L). It also fails to consistently highlight non-math questions that require quantitative reasoning, indicating that JE-IRT introduces additional ability-sensitive structure beyond the underlying encoder.
>
> -------
>
> >The cosine alignment between benchmark means correlates with OOD accuracy drops. Could this be formalized as a predictive diagnostic tool for estimating when a model will fail on a new dataset?
>
> We currently use cosine alignment only as a descriptive correlation rather than as a calibrated predictor. The mean direction of a benchmark is a very lossy summary of its geometry. More formally, let $\{E_{Q_j}\}$ be the question embeddings from the benchmarks used to fit JE-IRT, and define their minimal conical hull (convex cone) as
> $
> C_\text{seen} = \Bigl\\{\sum_j \alpha_j E_{Q_j} : \alpha_j \ge 0 \Bigr\\}.
> $
>
> A new benchmark $B$ is better represented by a distribution $\mu_B$ over its (unit-normalized) question embeddings on the sphere, not by a single mean vector. In an idealized geometric picture, the “in-distribution” mass of $B$ is the measure $\mu_B(C_{\text{seen}} \cap \mathbb{S}^{d-1})$: the fraction of its questions whose directions lie in the cone spanned by the seen data. Out-of-distribution performance would then be governed by an integral of the form
>
> $
>   \int_{C_{\text{seen}}} \ell(v, q)\, d\mu_B(q)
>   \quad \text{versus} \quad
>   \int_{\mathbb{S}^{d-1} \setminus C_{\text{seen}}} \ell(v, q)\, d\mu_B(q),
> $
>
> rather than by a single cosine between benchmark means.
>
> In practice, estimating $C_{\text{seen}}$ and $\mu_B$ well enough to turn this into a reliable diagnostic is computationally and statistically challenging in high dimension. Cosine alignment between benchmark means is therefore best viewed as a coarse proxy, not a standalone predictor. By contrast, JE-IRT already provides a practical predictive tool at the item level: for a new dataset we embed its questions, use the learned model vectors to predict each item's correctness probability, and aggregate if a benchmark-level score is needed. Designing lightweight cone-based diagnostics that approximate the integral above is an interesting direction for future work, but beyond the scope of the current paper.

---

### Comment · Area_Chair_SFqD · 2025-11-27
**Discussion Reminder to Reviewers**

Dear Reviewers,

The authors have responded to your reviews. Please engage in the discussion and evaluate the authors’ rebuttal to check whether your comments have been adequately addressed, and determine whether you would like to adjust your evaluations.

Best,

Your AC

---

### Author Response · Authors · 2025-11-30
**Summary of Contributions and Responses to Reviewer Concerns**

We thank the reviewers and AC for their careful evaluation. Below we summarize our contributions and how the revised manuscript addresses the main concerns.

**Summary of contributions.**
We introduce JE-IRT, a joint-embedding generalization of item response theory in which both questions and LLMs live in a shared low-dimensional space. We show that JE-IRT provides a principled geometric lens on how knowledge and abilities are organized by LLMs themselves, rather than being constrained by human-designed subject priors.

We summarize below the **main clarifications** made in response to the reviewers’ comments.

- **Relation to traditional IRT and novelty.**
The main difference from traditional IRT is that classical models do not generalize to unseen questions, whereas our formulation does by grounding question embeddings in their semantics. We add Appendix J to detail this theoretical connection and to compare JE-IRT and MIRT empirically in an overfitting regime.

- **Interpretability.**
Our framework primarily focuses on geometric interpretability. To enhance semantic interpretability, we add Appendix~L, where we use a DiffMean contrast to identify an arithmetic reasoning axis in the JE-IRT space and qualitatively study the questions that load highly on this axis.

- **Non-binary supervision and task formats.**
We add Appendix K, where we both generalize our formulation theoretically and empirically to  show that it is compatible with probability targets.

- **Generalization and sample efficiency.**
In Section 4.3, we show that our framework is data-efficient for onboarding new LLMs. In response to the reviewer’s question about adaptation, we added experiments for onboarding new benchmarks, demonstrating that a small subset can efficiently reduce the generalization gap.

- **Other clarifications.** We clarify our design choices and implementation details, sample efficiency and compute time, encoder dependence, and JE-IRT as a geometric foundation for future routing or adaptive-testing methods.

We appreciate the reviewers’ feedback, which inspired refinements that we hope meaningfully improve the clarity and accessibility of the paper.

---

### Meta-Review · Area_Chair_Vic4 · 2026-01-05

**Summary:**

The paper proposes JE-IRT, a novel geometric framework that extends Item Response Theory (IRT) by embedding both Large Language Models (LLMs) and evaluation questions into a shared vector space. Unlike traditional IRT, which relies on scalar parameters, JE-IRT uses the direction of question embeddings to represent semantic ability profiles and the norm to represent difficulty. Correctness is modeled as the dot product interaction between the model and question vectors.

The main concerns from reviewers about the novelty relevant to the similar work Multidimensional IRT (MIRT) and how this method is used to multiple-choice questions and free-generation settings.

**Reviewer Concerns:**

Addressed concerns:
- Regarding the computational cost question, the authors provided specific details (2 hours for full training, 20 minutes to onboard a new LLM), confirming the framework's efficiency
- For novelty compared to traditional IRT and MIRT, authors clarified that while MIRT uses lookup tables tied to specific questions (and thus cannot generalize to unseen items), JE-IRT uses a content-based encoder.

Outstanding concerns:
- Authors added all new results into appendix rather than the another page in the main body text, which seems not follow the restriction of ICLR rebuttal, as the new results added in the appendix are more than 5 pages, which cannot fit into the single page of the main body
- Although authors provide more experiments for non-binary test, a large-scale evaluation and analysis on purely generative/free-form tasks remains unclear

**Reviewer Scores:**

Apart from the reviewer 25eA's concerns being partially addressed, the remaining concerns about how this method generalizes to non-binary and free-form generation settings are still unclear, which tends to make review score of another two reviewers unchanged.

---

### Decision · Program_Chairs · 2026-01-26

Reject